



# 1 To what extents do urbanization and air pollution affect fog?

Shuqi Yan[1,2,3,4], Bin Zhu[1,2,3,4,*], Yong Huang[5,6], Jun Zhu[7], Hanqing Kang[1,2,3,4], Chunsong Lu[1,2,3,4], Tong Zhu[8]
[1]Collaborative Innovation Centre on Forecast and Evaluation of Meteorological Disasters, Nanjing University of Information
Science & Technology, Nanjing, China
[2]Key Laboratory for Aerosol-Cloud-Precipitation of China Meteorological Administration, Nanjing University of Information
Science & Technology, Nanjing, China
[3]Key Laboratory of Meteorological Disaster, Ministry of Education (KLME), Nanjing University of Information Science &
Technology, Nanjing, China
[4]Special test field of National Integrated meteorological observation, Nanjing University of Information Science & Tech-
nology, Nanjing, China
[5]Anhui Meteorology Institute, Key Lab of Atmospheric Science and Remote Sensing Anhui Province, Hefei 230031, China
[6]Shouxian National Climatology Observatory, Shouxian 232200, China
[7]Xiangshan Meteorological Bureau, Xiangshan 315700, China
[8]IMSG at NOAA/NESDIS/STAR, 5830 University Research Ct., College Park, MD 20740, USA
*Correspondence to*: Bin Zhu (binzhu@nuist.edu.cn)
**Abstract.** The remarkable development of China has resulted in rapid urbanization (urban heat island and dry island) and
severe air pollution (aerosol pollution). Previous studies demonstrate that these two factors have either suppressing or pro-
moting effects on fog, but what are the extents of their individual and combined effects? In this study, a dense radiation fog
event in East China in January 2017 was reproduced by the WRF-Chem model, and the individual and combined effects of
urbanization and aerosols on fog (indicated by liquid water content (LWC)) are quantitatively revealed. Results show that
urbanization inhibits low-level fog, delays its formation and advances its dissipation due to higher temperatures and lower
saturations. In contrast, upper-level fog could be enhanced because of the updraft-induced vapour convergence. Aerosols
promote fog by increasing LWC, increasing droplet concentration and decreasing droplet effective radius. Further experi-
ments show that the current pollution level in China is still below the critical aerosol concentration that suppresses fog. Ur-
banization influences fog to a larger extent than do aerosols. When urbanization and aerosol pollution are combined, the
much weaker aerosol promoting effect is counteracted by the stronger urbanization suppressing effect on fog. Budget analy-
sis of LWC reveals that urban development (urbanization and aerosols) alters LWC profile and fog structure mainly by mod-
ulating condensation/evaporation process. Our results infer that urban fog will be further reduced if urbanization keeps de-
veloping and air quality keeps deteriorating in the future.



# 1 Introduction

During the past five decades, China has achieved remarkable developments, accompanied by strong anthropogenic activities (rapid urbanization and severe air pollution). Urbanization and air pollution have significantly affected climate change, monsoons, air quality, fog, clouds and precipitation (e.g., Li et al., 2016; Li et al., 2017). Many studies have linked the changes in clouds and precipitation to urbanization and aerosols. Urbanization destabilizes the boundary layer, which triggers strong updrafts and invigorates convection (e.g., Rozoff et al., 2003; Shepherd, 2005). Aerosols modify the macroscopic, microphysics, thermodynamics and radiative properties of clouds through complicated pathways, which are called as aerosol-cloud-radiation interactions and have been systematically reviewed by Fan et al. (2016), Rosenfeld et al. (2014), Tao et al. (2012), etc. Fog can be viewed as a cloud (Leng et al., 2014) that occurs near the surface. Land use features and aerosol properties may instantly affect fog, so fog is more sensitive to anthropogenic activities than other types of clouds are (Zhu and Guo, 2016). Many studies have analysed the effects of urbanization and aerosols on fog, mostly in segregated manners.

Urbanization is featured with urban heat island (UHI) and dry island (UDI) effects. The urban surface has a lower albedo, which reduces the reflected solar radiation and enhances heat storage. Urban expansion decreases the coverage of cropland, water bodies and forestland, which reduces the sources of water vapour. As a result, urban areas commonly experience higher temperatures and lower vapour contents. These conditions induce a lower supersaturation that is unfavourable for fog formation (Gu et al., 2019). In the long-term scale, urban fog days are reported to decrease significantly (e.g., Guo et al., 2016; LaDochy, 2005; Sachweh and Koepke, 1995; Shi et al., 2008; Yan et al., 2019). Although UHI and UDI inhibit near-surface fog, the upward motions can promote upper-level fog (Li et al., 2011; Niu et al., 2010b). Surface roughness and thermal circulation cause strong updrafts (Rozoff et al., 2003), which transfer water vapour aloft and cause wet island phenomenon in the upper-level (Kang et al., 2014). The fog at that altitude may be subsequently enhanced.

Aerosols exert sophisticated impacts on fog through direct (radiation) effects and indirect (microphysical) effects (Khain and Pinsky, 2018). Scattering aerosols block downwelling solar radiation in the daytime, thus delaying the dissipation and elongating the duration of fog (Shi et al., 2008; Maalick et al., 2016). Although they increase downwelling longwave radiation at night, scattering aerosols have negligible effects on the fog formation time (Stolaki et al., 2015; Maalick et al., 2016). The role of absorbing aerosols like BC on fog depends on its residence height. If BC resides above the fog layer, BC causes a dome effect (Ding et al., 2016) which blocks solar radiation and prevents the dissipation of fog (Bott, 1991). If BC resides within the fog layer, BC heats fog droplets and accelerates the dissipation of fog (Maalick et al., 2016). The aerosol indirect effect on cloud is addressed as one of the most uncertain factors in the IPCC report (IPCC, 2013). Aerosol concentration has a two-fold effect on fog, which is called as the boomerang pattern (Koren et al., 2008). Under saturation conditions, increasing aerosols commonly result in more CCNs. It promotes activation and condensation, yielding more but smaller droplets





and increasing cloud water content (Fan et al., 2018; Rosenfeld et al., 2008). These changes have two kinds of positive feedback on fog (Maalick et al., 2016): more droplets cause stronger radiative cooling at fog top and enhance condensation (Jia et al., 2018); smaller droplet size inhibits sedimentation and the depletion of cloud water (Zhang et al., 2014). However, if aerosol concentration exceeds a certain threshold, this promoting effect disappears (Quan et al., 2011) or even turns into a suppressing effect due to the strong vapour competition (Koren et al., 2008; Rangognio, 2009). Additionally, large-scale aerosol pollution can change weather patterns and affect large-scale fog formation conditions (Niu et al., 2010a). Ding et al. (2019) found that the dome effects of BC induce a land-sea thermal contrast and generate a cyclonic anomaly over coastal areas. This anomaly results in more vapor transported inland and strengthened advection-radiation fog.

Yan et al. (2019) analysed decadal trends of fog days and quantitatively proved that the inhibiting effects of urbanization outweigh the promoting effects of aerosols on fog during the mature urbanization stage. Their study inspires us to quantitatively comfirm the roles of urbanization and aerosols in a dense fog event by an online-coupled synoptic and air quality model, WRF-Chem. This event is a radiation fog event with weak synoptic forcing (detailed in Sect. 3.1), so the effects of urbanization and aerosols should be obvious. Determining the quantitative extents of urbanization effect, aerosol effect and their combined effect is an interesting topic, which has barely been studied previously to the best of our knowledge. This work facilitates the understanding of how anthropogenic activities affect the natural environment, fog (cloud) physics and aerosol-cloud interactions near the surface.

In this study, urbanization mainly refers to UHI and UDI associated with land use change and human activities, excluding the increasing aerosol pollution caused by urban expansion. Air pollution refers to aerosols and is indicated by anthropogenic emissions because aerosol concentration is highly proportional to emission intensity. Liquid water content (LWC) and cloud/fog droplet number concentration ($N_d$) are two important parameters representing fog intensity and visibility. Following previous studies (e.g., Ding et al., 2019; Gu et al., 2019; Jia et al., 2018; Maalick et al., 2016; Yang et al., 2018), we use LWC as the indicator of fog to reveal different characteristics of fog in different experiments. This study is organized as follows. The data, model and methods are described in Sect. 2. Section 3.1 overviews the fog event and provides preliminary evidence of how urban development affects fog. Section 3.2 evaluates the model performance. Sections 3.3 to 3.5 analyse the urbanization, aerosol and combined effects on fog. Section 3.6 discusses the rationality and reliability of the results.

## 2 Data, model and methods

### 2.1 Data

The first data are the hourly automatic weather station data from the Shouxian National Climate Observatory (SX; 32.4° N,



116.8° E, 23 m) that are used to evaluate the model performance. SX is a rural site surrounded by vast croplands and is ap-
proximately 30 km away from the nearest large city (Fig. 1b). The data include horizontal visibility, temperature, relative
humidity, wind direction and speed. The second data are the Himawari 8 satellite data that are used to represent fog area
(https://www.eorc.jaxa.jp/ptree/index.html). Fog area is mainly indicated by the albedo at three visible bands: red (band 3,
0.64 μm), green (band 2, 0.51 μm) and blue (band 1, 0.47 μm). The third data are the 3-hourly data from the Meteorological
Information Comprehensive Analysis and Process System (MICAPS) (Li et al., 2010) that are also used to represent the fog
area. The fourth data are the land use data from the Moderate Resolution Imaging Spectroradiometer Land Cover Type Ver-
sion 6 data (MCD12Q1; https://lpdaac.usgs.gov/products/mcd12q1v006) in the year of 2017, the same as the simulation pe-
riod. The data are resampled from 500 m to 30 arc-seconds (approximately 1 km) and replace the geological data of the WRF
model.

## 2.2   Model configuration

The model used in this study is the WRF-Chem (V3.9.1.1) model. It is an online-coupled mesoscale synoptic and air quality
model that considers the sophisticated interactions among various dynamic, physical and chemical processes (Chapman et al.,
2009; Fast et al., 2006). WRF or WRF-Chem has been successfully used in simulating fog events (Jia and Guo, 2012; Jia and
Guo, 2015; Jia et al., 2018) and exploring aerosol-cloud interactions (Fan et al., 2018). Two nest domains are set up (Fig. 1).
The d01 domain has a size of 217×223 grids and a resolution of 6 km, covering the entire fog area of this event (Fig. 2a).
The d02 domain has a size of 115×121 grids and a resolution of 2 km, covering SX and the adjacent areas. The land use data
are replaced by MCD12Q1 data, which represent the latest condition.
Fog simulation is highly sensitive to vertical grids (Gultepe et al., 2007). A fine vertical resolution with a proper lowest
model level can better resolve turbulences, thus yielding a reasonable fog structure (Yang et al., 2019). Here, 42 vertical lev-
els are established with the first five η values of 1.000, 0.999, 0.998, 0.997, 0.996. There are 25 levels below the boundary
layer (approximately 1500 m), and the lowest model level is approximately 8 m.
Fog simulation is also sensitive to physical schemes (Gu et al., 2019). Through numerous experiments, radiation, micro-
physics and boundary schemes are found to significantly influence the model performance, and the boundary layer scheme
plays a decisive role (Naira Chaouch et al., 2017). The radiation schemes are the RRTM longwave scheme and the Goddard
shortwave scheme. The microphysical scheme is the Morrison double-moment scheme (Morrison et al., 2005). The boundary
layer scheme is the YSU 1.5-order closure non-local scheme, which yields better results than do any other schemes. The
major schemes are listed in Tab. 1.
The model is driven by the highest resolution product (0.125°, approximately 13 km) of ECMWF data



(https://apps.ecmwf.int/datasets/data/interim-full-daily/levtype=sfc/). The anthropogenic emissions are derived from the
Multi-resolution Emission Inventory for China (MEIC) database (http://www.meicmodel.org). The simulation starts at
2017-01-01 08:00 and ends at 2017-01-03 14:00, with the first 24 hours as the spin-up period (all the times here are in local
time).

## 2.3  Sensitivity experiments

The study site is SX because only its visibility is observed hourly and is a multiple of 1 m, which is suitable for evaluating
the model performance. To investigate the effects of urbanization and aerosols on fog, we change the land use and emission
intensity around SX. Four experiments, i.e., u0e0, u3e0, u0e3 and u3e3 are designed. The u0e0 is the base experiment, with
no urbanization and weak emission at SX. The u3e0 is set as the urbanization condition. The u0e3 is set as the polluted con-
dition. The u3e3 is set as the urban development condition (urbanization and pollution coexist). The experiment settings are
listed in Tab. 2.
On the setting of urbanized condition, we replace the land use of SX as that of Hefei, the most urbanized city and the capital
of Anhui Province. The downtown of Hefei has a built area of approximately 570 km$^2$. Therefore, the 11x13 box centered on
SX (572 km$^2$) is replaced by urban surface in the u3e0 and u3e3 experiments to represent the urbanization condition.
The downtown of Hefei has much higher emissions than SX. For example, the PM2.5 emission rate of Hefei is 40 times
higher than that of SX. To represent the polluted condition, the emission intensity of the aforementioned box is set to be
equal to that of downtown Hefei in the u0e3 and u3e3 experiments.

## 2.4  Calculating visibility

The LWC is the proxy of fog as mentioned above. Since the LWC is not observed, and visibility (VIS) is related to LWC, the
VIS is used to assess the model performance. VIS is not diagnosed by the model and can be parameterized by the function of
LWC, $N_d$ or droplet effective radius ($R_e$). Equation 1 (Kunkel, 1983) and 2 (Gultepe et al, 2006) are two parameterization
methods.

$$\text{VIS[m]}=27\text{LWC[g cm}^{-3}]^{-0.88} \tag{1}$$

$$\text{VIS[m]}=1002(\text{LWC[g cm}^{-3}]\cdot N_d[\text{cm}^{-3}])^{-0.6473} \tag{2}$$

Another parameterization method is based on the Mie theory (Gultepe et al., 2017). VIS is inverse proportional to atmos-
pheric extinction at visible wavelength. The extinction coefficient of cloud water ($\beta_c$) is



$$\beta_c\,\textbf{[km$^{-1}$]} = \frac{3Q_{ext}\,\rho_a\,\text{LWC}}{4\rho_w R_e} \times 10^6 \qquad\qquad (3)$$

where $\rho_a$ ($\rho_w$) is the air (water) density in kg m$^{-3}$, LWC is in g kg$^{-1}$, $R_e$ is in μm, and $Q_{ext}$ is the extinction efficiency, which is
assumed to be 2 for cloud droplets.
The atmospheric extinction ($\beta$) is also largely contributed by aerosols ($\beta_a$) and other types of hydrometeors. The model diag-
noses $\beta_a$ at 550 nm. No other types of hydrometeors occur in this fog case, so we assume $\beta = \beta_a + \beta_c$. Then VIS is determined
by the Koschmieder rule (Koschmieder, 1924): VIS[m]=3.912/$\beta$[km$^{-1}$]×1000.
During fog period (Fig. 4 shaded zone), the three methods nearly yield the same results (figure not shown), so the last meth-
od is used to calculate the simulated VIS.

# 3   Results and discussions

## 3.1   Overview of the fog event

### 3.1.1   Formation condition and lifetime

From 01 to 06 January 2017, East China is dominated by zonal circulation, with weak trough, ridge, pressure gradient and
atmospheric diffusion (Zhang and Ma, 2017). Under this stable weather pattern, the accumulation of pollutants and water
vapour promote the occurrence of fog-haze events. From the evening of 02 January to the noon of 03 January, a dense fog
event occurs in wide regions of East China. The fog reaches its peak at 08:00 03 January, covering south Hebei, east Henan,
west Shandong, Anhui, Jiangsu and Shanghai (Fig. 2a). Figure 4a shows the temporal variation of visibility at SX. The fog
forms at 18:00 02 January and dissipates at 12:40 03 January. This is a radiation fog which is promoted by strong radiative
cooling at night and weak easterly water vapour transport from northwest Pacific (Zhu et al., 2019).

### 3.1.2   Preliminary evidence of urban development affecting fog

Lee (1987) and Sachweh and Koepke (1995) observed "fog holes" over urban areas on satellite images. Here, fog hole means
the low liquid water path (LWP) region within the fog region, which is visualized as pixels with weak fog (high visibility) or
clear sky surrounded by dense fog. These holes demonstrate that urban development (urbanization and aerosols) has a clear-
ing effect on fog. In this fog event, fog holes are also present over urban areas on the Himawari 8 image at 11:00 03 January
(Fig. 3). We assume that urbanization and air pollution could have profound effects on fog by reducing the LWP or advanc-





ing the dissipation of fog.

## 3.2   Model evaluation and simulations

The model performance is evaluated by comparing the fog spatial coverage. Satellite cloud image and modelled LWP can
represent the observed and simulated fog zone, respectively (Jia et al., 2018). Figure 2 shows the Himawari 8 visible cloud
image and the simulated LWP distribution at 08:00. The light white pixels and light red dots indicate the observed fog area.
The model well captures the fog in south Hebei, east Henan, west Shandong, Anhui, Jiangsu and Shanghai.
The model performance is also evaluated by comparing the visibility and other basic parameters at the SX site (Fig. 4). Seen
from the visibility, the simulated fog forms at 19:30, 1.5 h later than the observation, and dissipates at 12:20, 30 min earlier
than the observation. During the fog period, the simulated visibility agrees well with the observation. The other parameters
such as temperature, wind speed and relative humidity are also effectively reproduced by the model, with relative small
RMSEs of 0.8 K, 0.7 m/s and 5.9 %, respectively. Overall, the model well captures the spatial feature and temporal evolution
of the fog.

## 3.3   Urbanization effects

From different sensitivity experiments (u3e0, u0e3 and u3e3), we can deduce the extents of the separate or combined effects
of urbanization and aerosols on fog. Figure 5 compares the LWC between u0e0 and u3e0. The general results are: (1) Before
02:00, urbanization leads to a decreasing LWC in all layers. Fog forms on the surface at 22:30 in u3e0, 3 h later than in u0e0.
(2) After 02:00, the LWC decreases in the low-level while it increases in the upper-level. Fog dissipates at 10:50 in u3e0, 1.5
h earlier than in u0e0. To better explain the LWC difference, its profiles are shown in Fig. 6. At 23:00, although fog has
formed in u3e0, the fog is rather weak compared with u0e0, which is caused by the higher temperature (Fig. 6f) and lower
saturation associated with UHI and UDI. At 02:00, fog develops in u3e0, but its intensity (the value of LWC) cannot reach
the same level as that in u0e0.
An interesting phenomenon is the opposite change of LWC in the low-level and upper-level after 02:00. This phenomenon
can be explained by the role of updrafts. The increasing roughness length and extra warming in urban conditions could trig-
ger horizontal wind convergence (Fig. S1) and the enhanced updrafts (Fig. 5c). The stronger updrafts in u3e0 affect conden-
sation via two possible pathways: (1) the vertical transport of vapour ($-w\frac{\partial q}{\partial z}$) and vertical convergence/divergence ($-q\frac{\partial w}{\partial z}$) re-
distribute water vapour and affect condensation; (2) the adiabatic cooling promotes condensation. The role of the first path-
way is measured by vertical vapour flux divergence ($\frac{1}{g}\frac{\partial(qw)}{\partial z}$). At 05:00, u3e0 shows a stronger vapour convergence above 110
m (Fig. 6h), and the LWC increases above 130 m (Fig. 6c). At 08:00, u3e0 shows a stronger vapour convergence above 130



m (Fig. 6i), and the LWC increases above 170 m (Fig. 6d). Therefore, it is possible that the adiabatic cooling and up-
draft-induced vapour flux convergence increase the vapour content and promote condensation in the upper-level, while the
fog in the low-level is suppressed by the divergence of vapour flux. At 11:00, fog disappears at the ground in u3e0 likely due
to the higher temperature (Fig. 6j). In summary, the UHI, UDI and updrafts alter the profile of LWC and reduce the LWP
most of the time (Fig. 5c), and the decreasing LWP in the daytime can explain why fog holes occur above urban areas (Fig.

198    3).

## 3.4    Aerosol effects

Figure 7 compares the LWC between u0e0 and u0e3. The formation time, dissipation time of fog and fog top show almost no
changes. The LWC increases at almost all layers in the polluted condition. Accordingly, the LWP also increases (Fig. 7c). It
is probable that the current pollution level of China always promotes fog occurrence. To testify whether the u0e3 is below
the transition point of the boomerang pattern, eight additional experiments (D10, D7.5, D5, D2.5, M2.5, M5, M7.5 and M10)
are performed. These experiments are the same as u0e3, except that the emissions around SX (the black box in Fig. 1b) are
multiplied (the "M" prefix) or divided (the "D" prefix). For example, the name M2.5 means multiplying by 2.5 times; the
name D10 means dividing by 10 times.
Figure 8 compares the LWC, $N_d$, $R_e$ and LWP among the nine emission-variant experiments. All the four parameters show
the boomerang pattern, which demonstrates that the model is able to simulate the dual effects of aerosols. Below u0e3, the
four parameters monotonically vary with emission level, indicating that aerosol pollution could always promote fog. This
phenomenon is because stronger emissions produce more aerosols and CCN. Under saturation conditions, the larger amount
of CCN boost activation and yield a higher $N_d$. The higher $N_d$ reduces $R_e$ and inhibits autoconversion and sedimentation
(Twomey, 1977); thus, this situation decreases the depletion of fog water and increases the LWC. This promoting effect has
been confirmed by many model studies (e.g., Maalick et al., 2016; Stolaki et al., 2015) and observations (e.g., Chen et al.,
2012; Goren and Rosenfeld, 2012). The aerosol concentration of the transition point (experiment M2.5) is higher than that of
u0e3 (Fig. 8), revealing that the current pollution level in China is still located in the promoting regime rather than the sup-
pressing regime of fog occurrence, which is also found by Jia et al. (2018).

## 3.5    Combined effects of urbanization and aerosols

Figure 9 compares the LWC between u0e0 and u3e3. The u3e3-induced change is quite similar to but not the same as the
u3e0-induced change. The time-height average of absolute change of LWC induced by u3e0, u0e3 and u3e3 are 0.120, 0.019,
0.124 g kg$^{-1}$, respectively. This result indicates that urbanization affects fog to a larger extent than do aerosols; when urbani-
zation and aerosols are combined, the effect of aerosols is indiscernible. The LWP is also significantly suppressed in the day-



time, and the promoting effect of aerosols in Fig. 7c is indiscernible in Fig. 9c. To further explain the changes in LWC, we
perform budget analysis of the LWC to determine which physical processes are the dominant contributors.
In WRF, the budget of LWC is composed of the following items,

$$\frac{\partial q_c}{\partial t} = \underbrace{-\left(u\frac{\partial}{\partial x} + v\frac{\partial}{\partial y} + w\frac{\partial}{\partial z}\right)q_c}_{\text{adv}} + \left(\frac{\partial q_c}{\partial t}\right)_{\text{PBL}} + \left(\frac{\partial q_c}{\partial t}\right)_{\text{micro}} + \left(\frac{\partial q_c}{\partial t}\right)_{\text{cumu}} \qquad (4)$$

where $q_c$ is LWC, and the subscripts denote advection, boundary layer, microphysical and cumulus processes, respectively.
The microphysical tendency is further decomposed into the following items,

$$\left(\frac{\partial q_c}{\partial t}\right)_{\text{micro}} = \left(\frac{\partial q_c}{\partial t}\right)_{\text{cold}} + \left(\frac{\partial q_c}{\partial t}\right)_{\text{auto}} + \left(\frac{\partial q_c}{\partial t}\right)_{\text{accr}} + \left(\frac{\partial q_c}{\partial t}\right)_{\text{sedi}} + \left(\frac{\partial q_c}{\partial t}\right)_{\text{cond/evap}} \qquad (5)$$

where the subscripts denote cold phase processes, autoconversion, accretion, sedimentation and condensation/evaporation,
respectively.
All the processes regarding precipitation and cold phase (the cumu, cold, auto and accr subscripts) are not analysed because
no precipitation occurs, and the temperature is above $0^{\circ}$C in the simulated fog (figure not shown). The sum of microphysical
(condensation/evaporation and sedimentation), boundary layer and advection tendencies is equal to the LWC distribution, so
the contributions of other physical processes can be safely ignored.
We can also infer that to what extents the various physical processes affect fog through the sensitivity experiments (u3e0,
u0e3 and u3e3). Additional aerosols weakly influence these processes (Fig. S2 right column) and subsequently result in weak
LWC change (Fig. 7c). Compared with aerosols, urbanization effect is much more considerable (Fig. S3 right column); it
dominantly accounts for the variation in physical tendencies from u0e0 to u3e3 (Fig. 10 right column). In u3e3 condition,
urban development (urbanization and aerosols) induces different magnitude of changes in different physical tendencies. The
relative magnitudes are 52.1, 38.3 and 9.6 % for the microphysical, boundary layer and advection processes, respectively,
indicating that microphysics is most susceptible to urban development and contributes most to the LWC change. Among
various microphysical processes, condensation/evaporation contributes most (72.7 %) to the change in microphysical ten-
dency (Fig. 11 right column). The above results indicate that urban development affects the LWC mainly by modulating the
condensation/evaporation process. Since u3e3 condition still witnesses higher temperatures and stronger updrafts (figure not
shown), the notable variation in condensation/evaporation tendency induced by u3e3 can also be attributed to the predomi-
nant role of UHI, UDI and updrafts. The mechanism has been analysed in Sect. 3.3.





## 3.6  Discussions

As mentioned above, urbanization influences fog to a larger extent than do aerosols; the LWC in fog does not vary substantially with pollution level. This section discusses the rationality and reliability of our results through mechanism analysis and observational evidence.

The sensitivity of cloud properties to aerosols depends on aerosol concentration and saturation environment. In convective clouds with intense upward motions and high saturations, the response of cloud properties to additional aerosols is significant ("aerosol-limited regime") (Fan et al., 2018). However, in fog with much weaker updrafts and lower saturations, this response could be more sensitive to vapour content rather than aerosol concentration ("vapour-limited regime"). It possibly implies that the LWC in fog varies slightly with pollution level but considerably with saturation condition that related to urbanization. Our results reveal that the time-height average LWC varies within the extent of 0.07g kg$^{-1}$ when emission intensity varies within two orders of magnitude (Fig. 8). This relative weak response of the LWC to pollution level is also reported by Jia et al. (2018).

In terms of observational evidence, Yan et al. (2019) revealed that fog days in polluted regions of East China have decreased since the 1990s. Through quantitative analysis, the promoting effects of aerosols are weakening, while the suppressing effects of urbanization are enhancing and dominantly cause this decrease. Sachweh and Koepke (1995) also claimed that the hindering effects of urbanization outweigh the promoting effects of aerosols on fog in southern Germany. Additionally, satellite images present discernible fog holes above urban areas (Fig. 3) (Lee, 1987; Sachweh and Koepke, 1995). Therefore, these observational evidence support the model results that the promoting effect of aerosols is counteracted by the hindering effect of urbanization. We believe that the results can also be applied to other cities in China because these cities commonly witness strong UHI, UDI and severe air pollution.

# 4  Conclusions

A dense radiation fog event occurred in East China from 02 to 03 January 2017. Satellite images show that fog holes occur over urban areas, demonstrating the remarkable effects of urbanization and air pollution on fog. Hence, the mechanism is investigated by the WRF-Chem model. The model well captures the spatial coverage and temporal evolution of the fog. Furthermore, the separate and combined effects of urbanization (refers to UHI and UDI) and air pollution (refers to aerosols) on fog (indicated by the LWC) are revealed, and the extents of these effects are quantitatively determined. Results show that:

Urbanization redistributes the LWC profile by the UHI, UDI effect and updrafts. The updrafts may be caused by surface





roughness and extra warming. The UHI and UDI suppress low-level fog, delay its formation by 3 h, and advance its dissipation by 1.5 h. However, the upper-level fog could be enhanced due to the updraft-induced adiabatic cooling and vapour flux convergence. Urbanization reduces the LWP most of the time, and this reduction in the daytime can explain why fog holes are present above urban areas on satellite images.

Aerosols promote fog mainly by changing microphysical properties. The increasing emissions (aerosol concentration) produce more CCN and fog droplets, which decreases $R_e$ and inhibits sedimentation, thus leading to a higher LWC. Further sensitivity experiments show that the current pollution level in China is still below the transition point of the boomerang pattern that suppresses fog. The macroscopic properties such as fog top and lifetime remain nearly unchanged.

The role of urbanization far overweighs that of aerosols. Therefore, when they act together, the urbanization effect is dominant, and the aerosol effect is indiscernible. Budget analysis of LWC shows that increasing aerosols influence various physical processes to a lesser extent, while urbanization influences these processes to a larger extent, eventually leading to a substantial LWC change in urban development condition (urbanization and aerosols). In this condition, comparisons among various physical processes reveal that microphysics dominates the change in LWC, and condensation/evaporation dominates the change in microphysical tendency. This result highlights the importance of condensation/evaporation process in modulating the LWC profile and fog structure.

Mechanism analysis and the observational evidence support our key finding that urbanization influences fog to a much larger extent than do aerosol pollution. Therefore, we believe our results are reasonable and robust in radiation fog events without strong synoptic forcings, and the results can also be applied to other cities in China due to the similar urban development patterns. This study facilitates a better understanding of how anthropogenic activities affect the natural environment, fog (cloud) physics and aerosol-cloud interactions near the surface. We can also infer the future change of fog occurrence. Under the traditional urban development pattern, i.e., urbanization keeps developing and air quality keeps deteriorating, urban fog occurrence will be further reduced.

*Code and data availability*. Some of the data repositories have been listed in Sect. 2. The other data, model outputs and codes can be accessed by contacting Bin Zhu via binzhu@nuist.edu.cn.

*Author contributions*. SY performed the model simulation, data analysis and manuscript writing. BZ proposed the idea, supervised this work and revised the manuscript. YH provided the observation data at the SX site. JZ processed the observation data. HK offered helps to the model simulation. CL and TZ also contributed to the manuscript revision.




*Competing interests.* The authors declare that they have no conflict of interest.

*Acknowledgments.* We are grateful to the High Performance Computing Center of Nanjing University of Information Science
and Technology for doing the numerical calculations in this work on its blade cluster system. We thank American Journal
Experts (AJE) for the English language editing.

*Financial support.* This work is supported by the National Key Research and Development Program (2016YFA0602003)
and the National Natural Science Foundation of China (91544229, 41575148, 41605091).

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





Table 1. Summary of major parameterization schemes.

| Scheme | Option |
|---|---|
| Boundary layer | YSU |
| Longwave radiation | RRTM |
| Shortwave radiation | New Goddard |
| Microphysics | Morrison |
| Surface layer | MM5 similarity |
| Land surface | Noah |
| Urban surface | Urban canopy model |
| Gas phase chemistry | CBMZ |
| Aerosol chemistry | MOSAIC (4-bin) |
| Aerosol-cloud-radiation interactions | All turned on |
| Aerosol activation | Abdul-Razzak and Ghan (2002) |







Table 2. Settings of sensitive experiments. "N" represents no changes.

| Case name | Description | Underlying surface | Anthropogenic emission |
|---|---|---|---|
| u0e0 | base condition | N | N |
| u3e0 | urbanization condition | the 11x13 grid centered on SX is replaced by urban surface | N |
| u0e3 | polluted condition | N | the 11x13 grid centered on SX is replaced by the emission of Hefei downtown |
| u3e3 | urbanization and polluted condition | same as u3e0 | same as u0e3 |

| Effect | Description |
|---|---|
| u3e0-u0e0 | urbanization effect |
| u0e3-u0e0 | aerosol effect |
| u3e3-u0e0 | urbanization and aerosol effect |






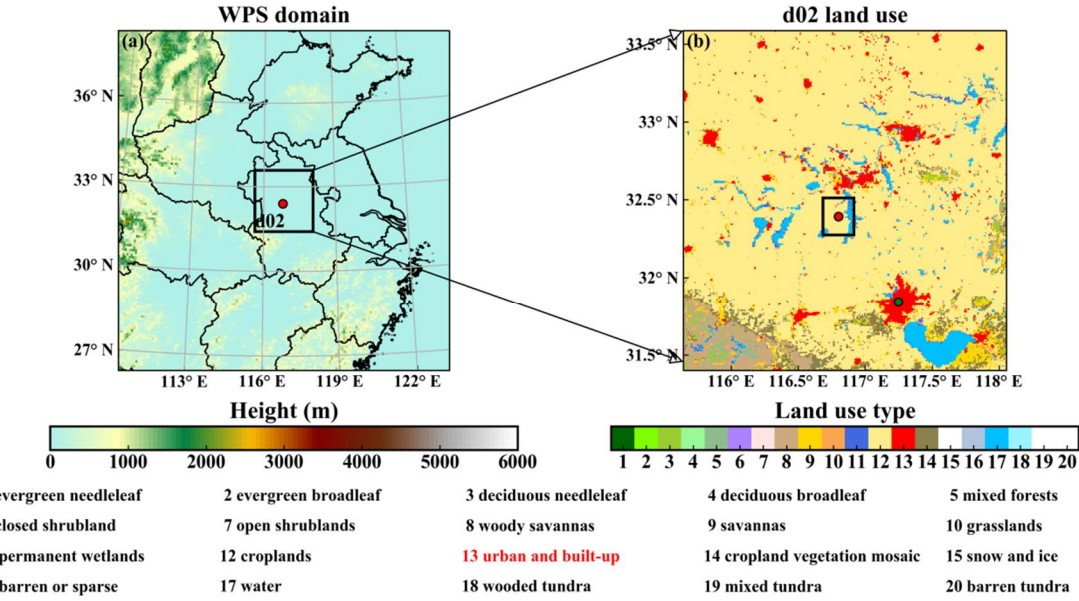

Figure 1. (a) The WRF domain overlaid with terrain height. (b) The land use distribution of domain d02. The green dot is Hefei, the capital of Anhui Province. The two red dots are the SX site. The land use and emissions of the 22 km × 26 km black box in the center of (b) will be altered in the sensitivity experiments.





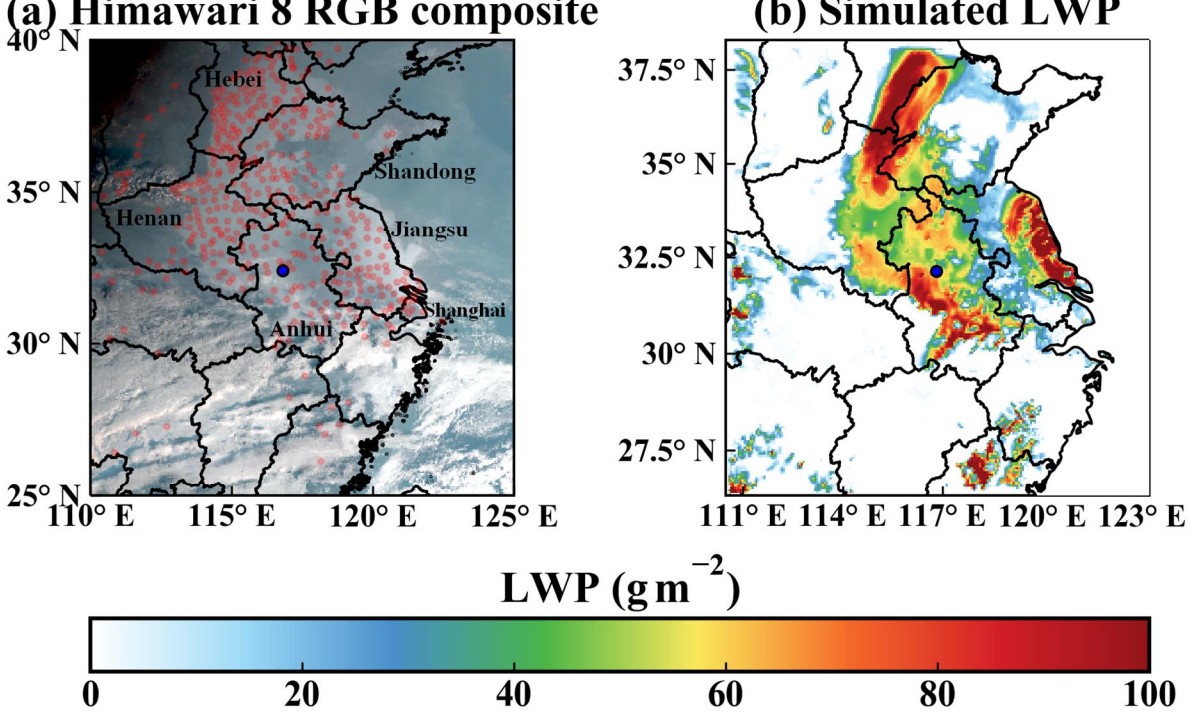


Figure 2. The performance of the simulated fog zone at 08:00 03 January. (a) Himawari 8 RGB composite cloud image
overlaid with the MICAPS observation sites (light red dots) at which fog was observed (relative humidity > 90 % and
VIS < 1 km). (b) Simulated LWP distribution. Only LWC below 1500 m are integrated. The blue dots are the SX site.



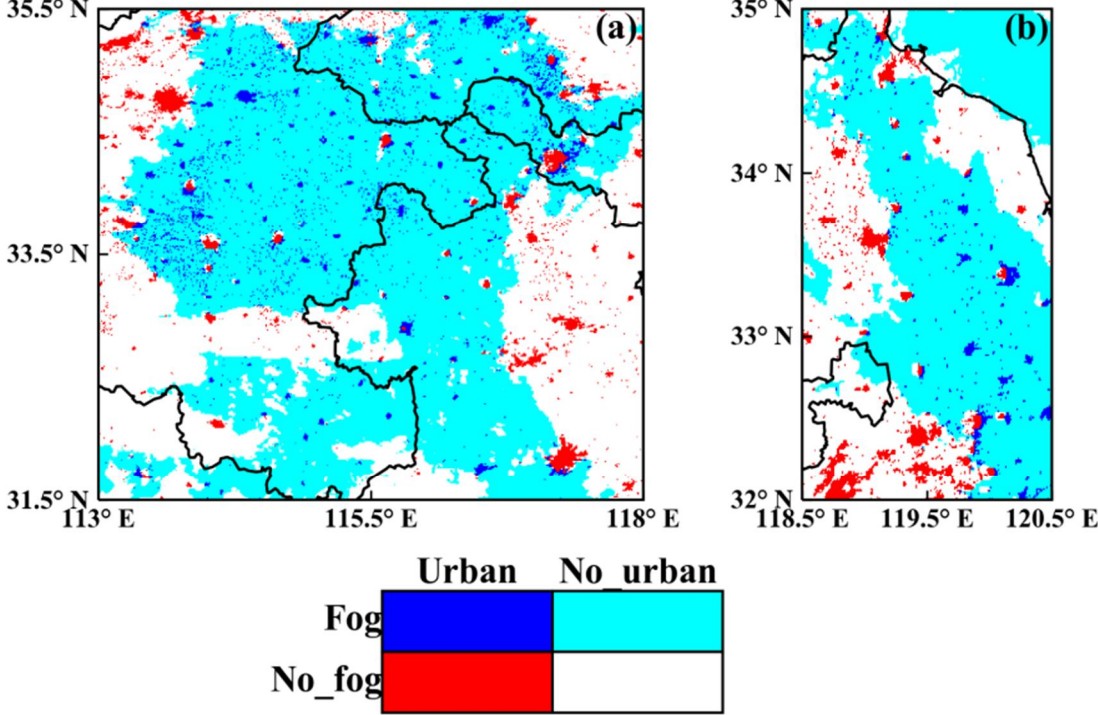


Figure 3. Two sub-regions (a and b) with obvious fog holes on the Himawari 8 image at 11:00 03 January. The fog zone, which is represented by albedo > 0.45 (at 0.64 μm) and brightness temperature > 266 K (at 12.4 μm) (Di Vittorio et al., 2002), is marked with cold colours (blue or cyan). The urban areas are marked with blue or red. The red and white pixels surrounded or semi-surrounded by cold colours are fog holes, and among these pixels, the red pixels indicate the fog holes over urban areas.





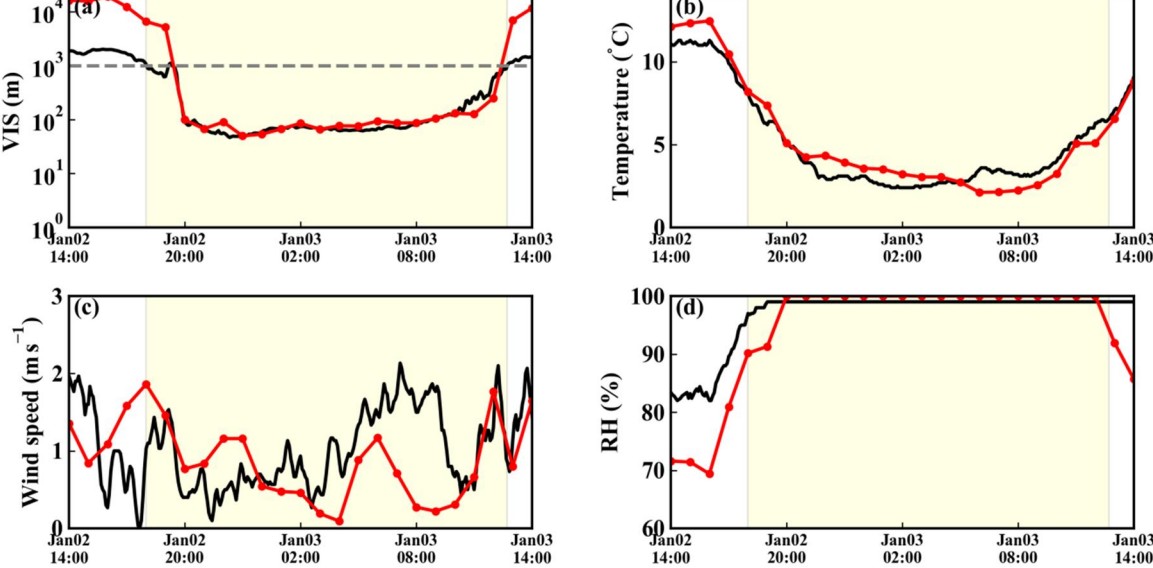


Figure 4. The performance of the simulated meteorological parameters at the SX site. (a) VIS. (b) air temperature. (c) 10-minute average wind speed. (d) Relative humidity (RH). The red dotted lines represent the model results, and the black lines are the observations. The fog period (VIS < 1 km and RH > 90 %) is shaded with light yellow.



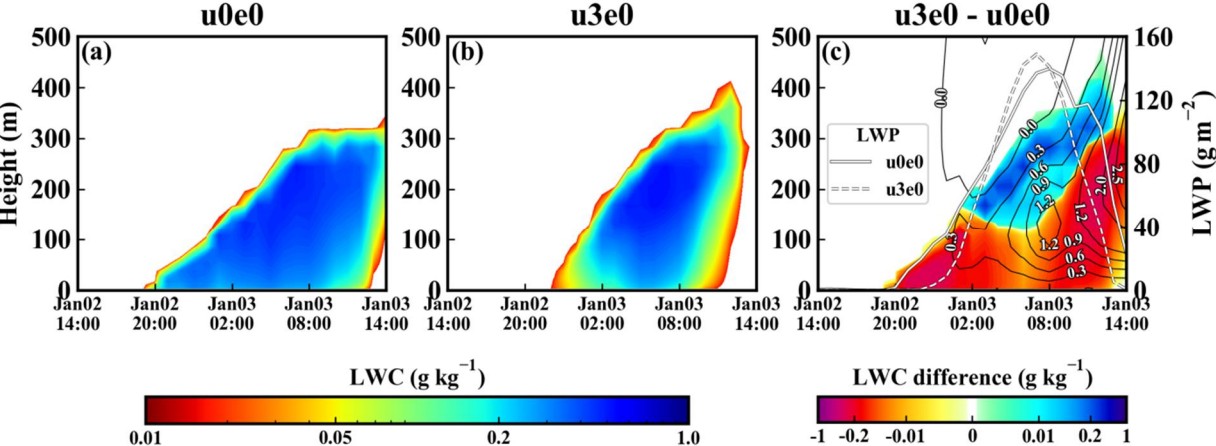


Figure 5. Time-height distribution of the LWC (g kg⁻¹) in (a) u0e0 and (b) u3e0, and (c) is the urbanization effect (u3e0 minus u0e0) on LWC. The two white curves in (c) are the LWP. The black contour lines in (c) are the difference of vertical velocity (cm s⁻¹) (u3e0 minus u0e0). Only the lines after 00:00 are shown for clarity.



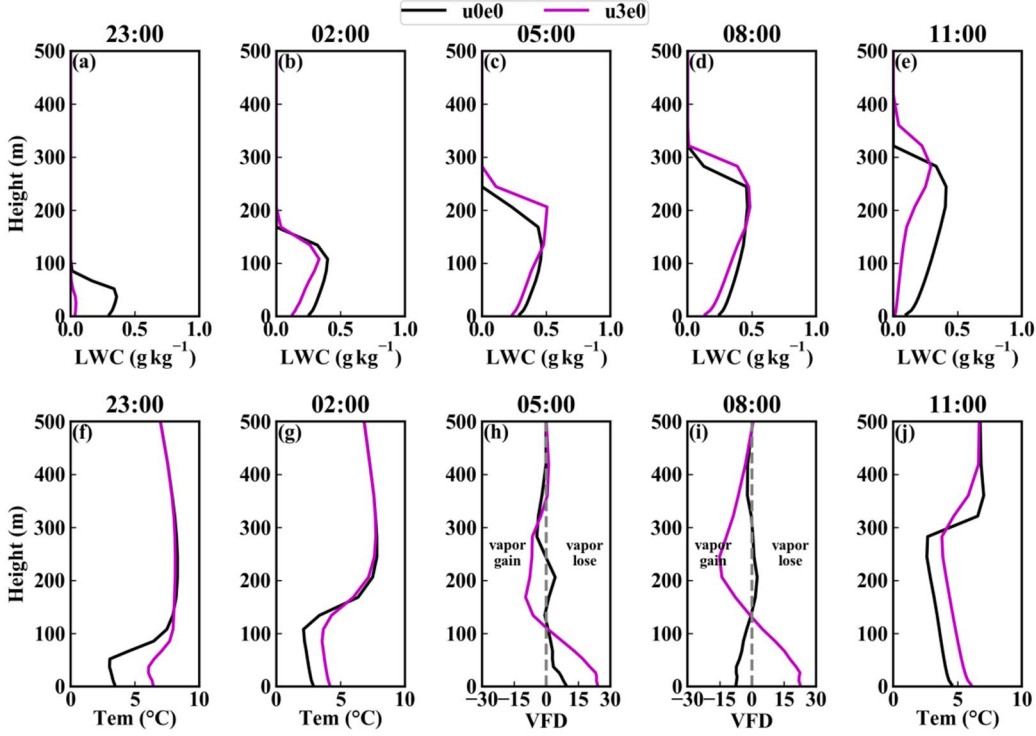


Figure 6. Profiles of the LWC (first row), temperature (Tem) (f, g, j) and vertical vapour flux divergence (VFD) (h, i)

(g h$^{-1}$ m$^{-2}$·hpa$^{-1}$) in u0e0 and u3e0 at different times.







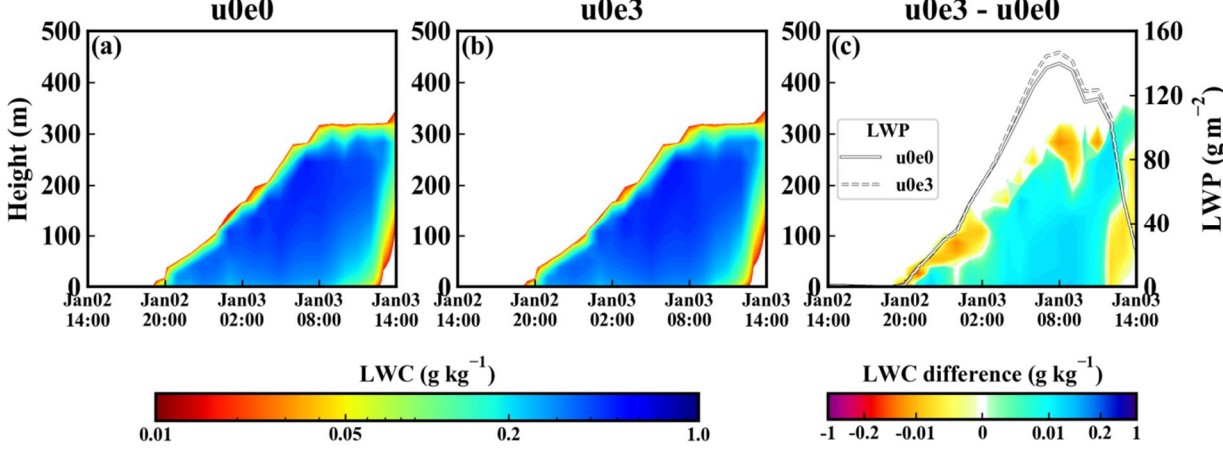


Figure 7. Similar to Fig. 5, but for the aerosol effect (u0e3 minus u0e0).




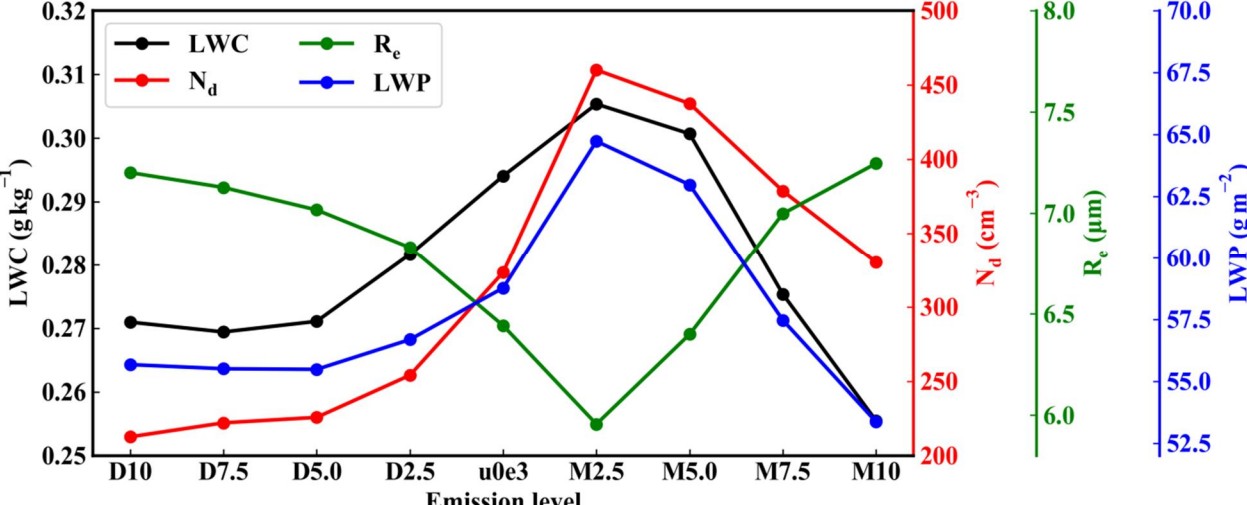


Figure 8. Relationships of the microphysical parameters (LWC, $N_d$, $R_e$ and LWP) with emission level. These parameters are the time-height averages (time average for the LWP), taking only non-zero values into consideration.





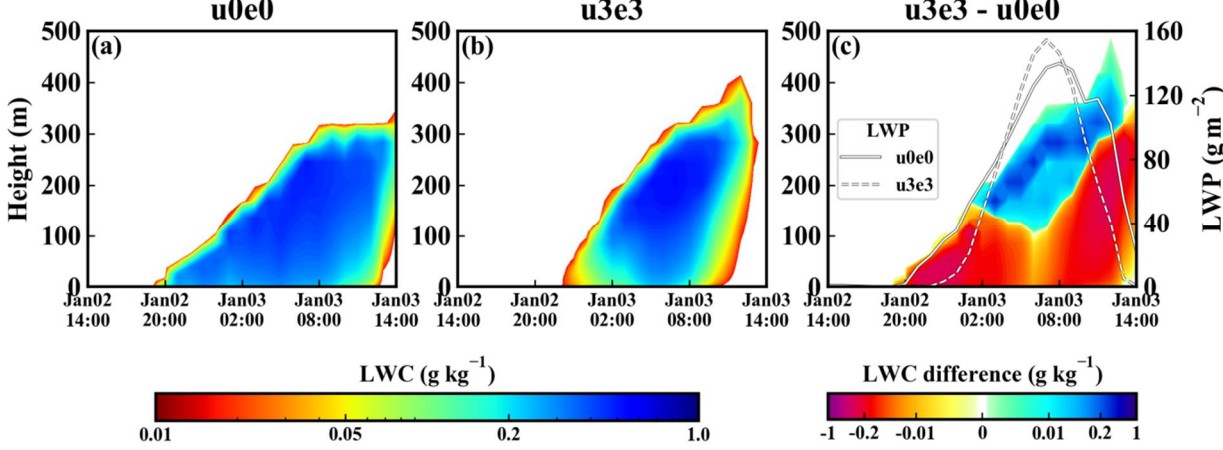


Figure 9. Similar to Fig. 5, but for the combined effect of urbanization and aerosols (u3e3 minus u0e0).




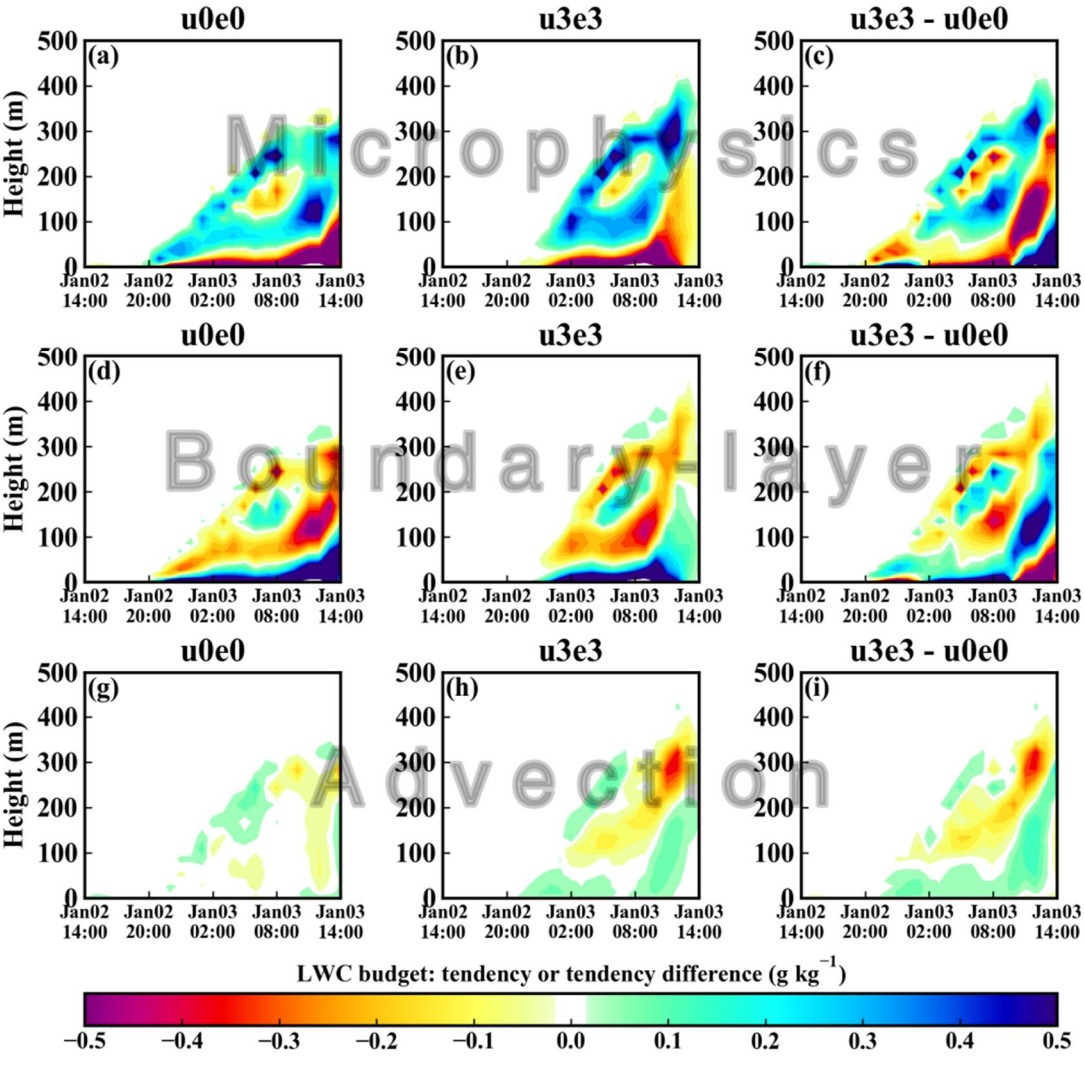


Figure 10. The combined effect of urbanization and aerosols (u3e3 minus u0e0) on various items of the LWC budget.
The three rows are the 1-hour accumulated tendencies (g kg$^{-1}$) of the microphysical, boundary layer, and advection
processes.



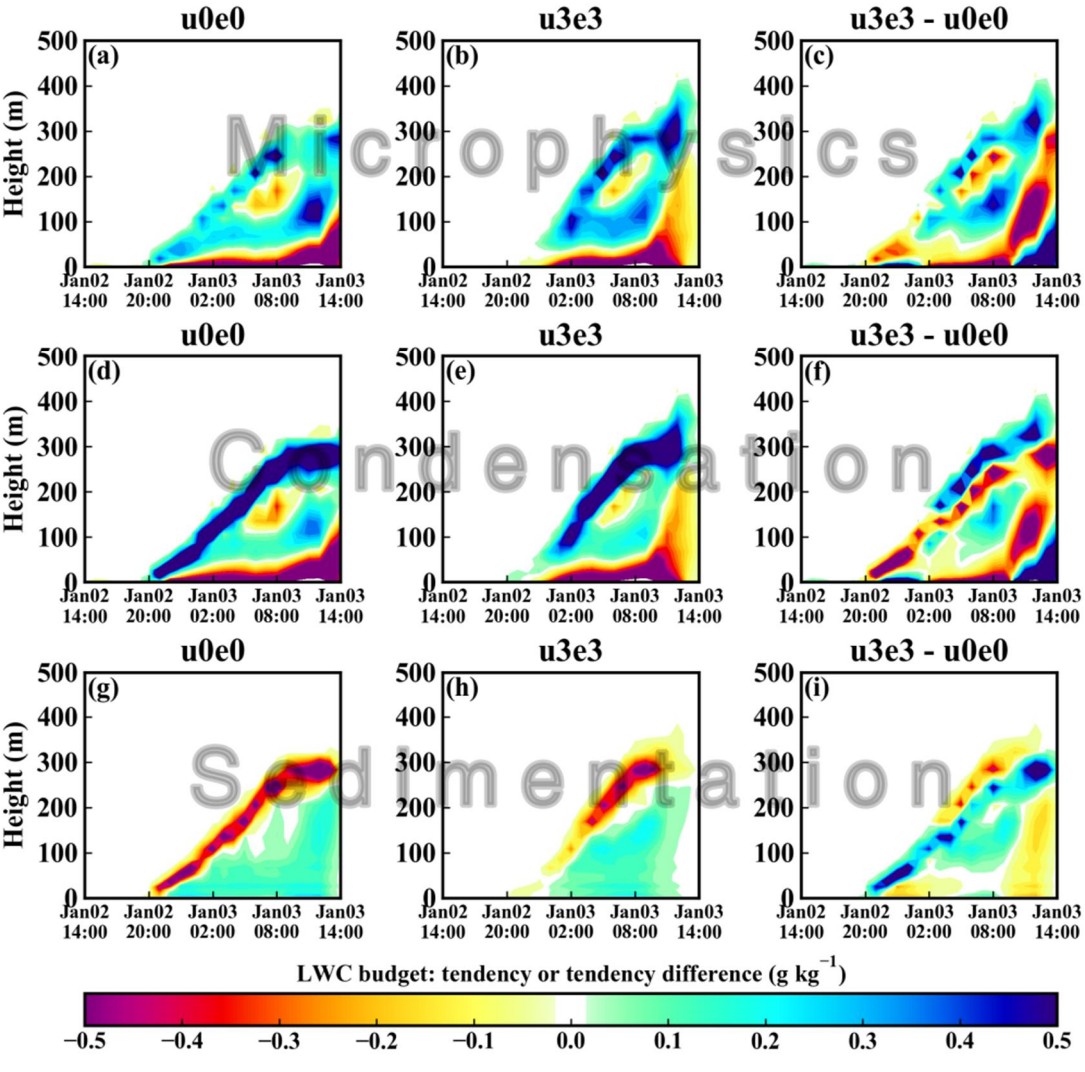


Figure 11. The combined effect of urbanization and aerosols (u3e3 minus u0e0) on various items of the microphysical
tendency. The three rows are the 1-hour accumulated tendencies (g kg$^{-1}$) of the microphysical, condensa-
tion/evaporation, and sedimentation processes.