# Peer review of "To what extents do urbanization and air pollution affect fog?"

_Atmospheric Chemistry and Physics, 2019_

## Referee Comment (RC1) · Anonymous Referee #3 · 27 Jan 2020

This paper investigates the effects of urbanization on the fog formation in eastern China. The Authors study the effects of two important factors on the fog formation, i.e., urbanization and aerosol particles. Using the WRF-Chem model, the contribution of the individual and the combination effects of these factors to the fog formation are estimated. The data and the model used in this study are reliable, and the analysis of the results is supported this study. Because the region of this study (eastern China) is under a rapid development stage, the causes on the environmental and ecosystem effects need to be carefully studied. The content of this study fits the scientific committee needs, and suitable for the publication in ACP. I have some comments for the paper, and the paper can be published after the Authors carefully address my follow-

ing comments. (1) The focus of this study is on the radiative fog. However, there are different fog formation in atmosphere. For example, the advection fog formation is often occurred in the coast of eastern China. The Authors should highlight that under different fog conditions (i.e., radiative fog or advection fog, etc.) what is the effects of the urbanization and aerosol particles on the fog formation. (2) Is this study suitable for the most of large cities in eastern China? (3) Some important references are missing. For example, Tie et al (2017) studied the important feedback of atmospheric moister on the aerosol pollution in eastern China, which should state in the instruction.

References

Tie, X., R.J. Huang, J.J. Cao, Q. Zhang, Y.F. Cheng, H. Su, D. Chang, U. Pöschl, T. Hoffmann, U. Dusek, G. H. Li, D. R. Worsnop, C. D. O'Dowd, Severe Pollution in China Amplified by Atmospheric Moisture, Sci. Rep. 7: 15760 | DOI:10.1038/s41598-017-15909-1, 2017. Tie, XX, X. Long, GH Li, SY Zhao, JJ Cao, JM Xu, Ozone enhancement due to photo-dissociation of nitrous acid in eastern China, Atmos. Chem. Phys., 19, 11267–11278, 2019.

―――――――――――――――――――――――――

---

## Referee Comment (RC2) · Anonymous Referee #1 · 29 Jan 2020

General comments:

The relative contribution by aerosol and urbanization to the fog formation is highly debated due to the competing but opposite effect. To tackle this challenge, this manuscript by Yan et al. attempted to quantitatively elucidate how urbanization and air pollution, both of which afflict the rapid economic countries like China and India, affect fog by means of sensitive model experiments. This topic is very interesting, and it is worth of further investigation. Overall, this manuscript is well written, and the methodologies are straightforward and easy to follow. However, the authors are recommended to adequately address the following concerns before consideration for acceptance for publication by ACP.

Specific comments:

1. Line 37-38: As indicated in IPCC AR5, "aerosol-cloud-radiation interactions" is suggested to be rephrased as "aerosol-radiation interaction" and "aerosol-cloud interaction" separately.

2. Line 41: "Many"->"Previous"

3. Line 45: "lower supersaturation" ??

4. Lines 64-65: Some important references are missing regarding the observational evidences of aerosol boomerang effect in China, e.g., Wang et al., AE 2015, doi: 10.1016/j.atmosenv.2015.04.063; Guo et al., GRL 2017, doi: 10.1002/2017GL073533; Liu et al., Sci. Rep. 2019, doi:10.1038/s41598-019-44284-2.

5. Lines 69-71: I notice that the work by Yan et al. JGR (2019) mentioned here is also from the same research group. Also, it occurs to me that the motivation seems a little confused: Since previous work has "quantitatively" proved…, why the authors attempt again to "quantitatively" confirm by model simulation of a fog event. Two "quantitatively" is redundant. Therefore, this sentence is suggested to be rephrased as follows: e.g. Our recent observational work (Yan et al., 2019) indicated a decreasing trend in fog days, and …"

6. Line 75: "facilitates"-> "is expected to facilitate"

7. Line 85: Something is suggested to be mentioned concerning Section 4 immediately after "Section 3.6 discusses the rationality and reliability of the results."

8. Line 90: it is suggested to clarify which city you are referring to? Since the reader cannot easily get any info from either text or Figure 1b.

9. Line 97: "replace"-> "used to replace"

10. Lines 160-165: The logic seems a little problematic: since the fog holes are mainly caused by urbanization, as demonstrated in the references in this paragraph (aerosol effect is not mentioned and is supposed to not be the focus here), why you mentioned the effect of aerosol pollution. It is generally thought that urbanization effect tends to reduce LWP whereas aerosol tends to accumulate the formation of fog. The combined effect is highly dependent on the competing effect of the two factors. Here it is not accurate to argue that both of them "reducing the LWP or advancing the dissipation of fog".

11. Section 3.2: What are the criteria for you to determine a fog event from model-simuated LWP, which is required to be clarified here.

12. Section 3.3: The authors attempted to discuss the complicated non-monotonic effect of aerosol on fog formation by differing the emission rate, which is not very common. Why not used the aerosol concentration or CCN or Na that can well represent the real atmospheric pollution level for the time period investigated here? I am curious of the actual CCN or Na concentration for the experiment of u0e3?

13. Lines 215-216: It will be misleading for the statement "the current pollution level in China is still located in the promoting regime rather than the suppressing regime of fog occurrence". Both ideal simulation (e.g., Rosenfeld et al. Science, 2008) or observational studies (Wang et al., AE 2015; Guo et al., GRL 2017) indicated that the tipping point tends to occur at AOD of 0.3-0.4 or CCN concentration of $1200/cm^3$. Recent observational work by Ilan Koren et al. (Science, 2014) suggested the cloud and precipitation is most sensitive to aerosol over the South Ocean. By comparison, the average AOD from MODIS in East China is on average much larger than 0.6, irrespective of the meteorological conditions.

14. Lines 205-206: times is redundant and should be removed.

15. Line 448: it is better to indicate the year of 2017 following 03 January in the figure caption.

16. Figure 3: it is suggested to show the major cities in the regions of interest shown in panel a and b, given the convenience to better understand the fog hole induced by urban. Besides, two subregions in Figure 3 is better to be marked in Figure 1 or 2.

---

## Referee Comment (RC3) · Anonymous Referee #3 · 11 Feb 2020

I think the authors answer my comments, and I don't have more comments. The paper can be accepted.

---

## Author Comment (AC1) · 11 Feb 2020

Dear Referees,

Thanks for giving us an opportunity to revise our manuscript (acp-2019-1045). We appreciate your positive and constructive comments. We have studied these comments carefully and make revisions on the manuscript. These comments and the corresponding replies are listed below.

The referee's comments are highlighted by gray, and followed by the comments are our responses. The symbol "≫" is the original texts in the manuscript. The modified sentences are marked by underlines. The added/replaced texts are colored by red.

With regards,

Shuqi Yan, Bin Zhu*, and all co-authors.

**Replies to Referee#3**

1. The focus of this study is on the radiative fog. However, there are different fog formation in atmosphere. For example, the advection fog formation is often occurred in the coast of eastern China. The Authors should highlight that under different fog conditions (i.e., radiative fog or advection fog, etc.) what is the effects of the urbanization and aerosol particles on the fog formation.

Thank you for this valuable suggestion. We agree that radiation fog and advection fog are two major fog types in China. They can occur in both inland and coastal areas. Gu et al. (2019) revealed the occurring frequencies of different fog types in Shanghai, a coastal city, during the past three decades. The major fog type is radiation fog (38.3%), followed by advection fog (27.7%) and advection-radiation fog (23.4%). Therefore, we infer that the dominant fog type in inland areas and coastal areas is radiation fog, which should be attracted more attention.

Compared with radiation fog which usually occurs under stagnant weather conditions, advection fog is associated with synoptic forcing, i.e., advection of a moist air mass with contrasting temperature properties with respect to the underlying surface (Gultepe et al., 2007). The role of synoptic forcing should be considered when studying the effects of urbanization and aerosols on advection fog, which is more complex than radiation fog. Zhong et al. (2017) indicated that urbanization and aerosols have nonsignificant effects on convective precipitation when the synoptic forcing is strong. Therefore, this study focuses on radiation fog to study the effects of urbanization and aerosols.

References

Gu, Y., Kusaka, H., van Doan, Q., and Tan, J.: Impacts of urban expansion on fog types in Shanghai, China: Numerical experiments by WRF model, Atmos. Res., 220, 57–74, https://doi.org/10.1016/j.atmosres.2018.12.026, 2019.

Zhong, S., Qian, Y., Zhao, C., Leung, R., Wang, H., Yang, B., Fan, J., Yan, H., Yang, X., and Liu, D.: Urbanization-induced urban heat island and aerosol effects on climate extremes in the Yangtze River Delta region of China, Atmos. Chem. Phys., 17, 5439–5457, https://doi.org/10.5194/acp-17-5439-2017, 2017.

2. Is this study suitable for the most of large cities in eastern China?

Thank you for this valuable suggestion. In the reply to comment1, we infer that the dominant fog type in inland areas and coastal areas of eastern China is radiation fog. Under the unified leadership of national government, most of the cities in eastern China experience the similar development pattern, i.e., the expansion of urban areas is commonly accompanied by increasing aerosol pollution. So we believe our results of radiation fog are suitable for most of the cities in eastern China.

3. Some important references are missing. For example, Tie et al (2017) studied the important feed-back of atmospheric moister on the aerosol pollution in eastern China, which should state in the instruction.

Thank you for this valuable suggestion. We have added the two references in Line 54-55: " Aerosols attenuate shortwave radiation, influencing PBL structure and the vertical profile of moisture and aerosols (Tie et al., 2017, 2019), which can alter the formation and dissipation condition of fog".

References

Tie, X., R.J. Huang, J.J. Cao, Q. Zhang, Y.F. Cheng, H. Su, D. Chang, U. Pöschl, T. Hoffmann, U. Dusek, G. H. Li, D. R. Worsnop, C. D. O'Dowd, Severe Pollution in China Amplified by Atmospheric Moisture, Sci. Rep. 7: 15760 DOI:10.1038/s41598-017-15909-1, 2017.

Tie, XX, X. Long, GH Li, SY Zhao, JJ Cao, JM Xu, Ozone enhancement due to photo-dissociation of nitrous acid in eastern China, Atmos. Chem. Phys., 19,11267–11278, 2019.

---

## Author Comment (AC2) · 11 Feb 2020

Dear Referees,

Thanks for giving us an opportunity to revise our manuscript (acp-2019-1045). We appreciate your positive and constructive comments. We have studied these comments carefully and make revisions on the manuscript. These comments and the corresponding replies are listed below.

The referee's comments are highlighted by gray, and followed by the comments are our responses. The symbol "≫" is the original texts in the manuscript. The modified sentences are marked by underlines. The added/replaced texts are colored by red.

With regards,

Shuqi Yan, Bin Zhu*, and all co-authors.

**Replies to Referee#1**

1. Line 37-38: As indicated in IPCC AR5, "aerosol-cloud-radiation interactions" is suggested to be rephrased as "aerosol-radiation interaction" and "aerosol-cloud interaction" separately.

≫Line 37-38: ...which are called as aerosol-cloud-radiation interactions...

Thank you for this valuable suggestion. We have rephrased it to be "...which are called as aerosol-radiation and aerosol-cloud interactions..."(Line 38-39).

2. Line 41: "Many"->"Previous"

≫Line 41: Many studies have analysed...

Thank you for this valuable suggestion. We have changed all the "many studies" to "previous studies" (Line 35, 42, 219).

3. Line 45: "lower supersaturation" ??

≫Line 45: As a result, urban areas commonly experience higher temperatures and lower vapour contents. These conditions induce a lower supersaturation that is unfavourable for fog formation.

Thank you for this valuable suggestion. We change this sentence to "…These conditions induce a lower relative humidity supersaturation that is unfavourable for fog formation" (Line 47).

4. Lines 64-65: Some important references are missing regarding the observational evidences of aerosol boomerang effect in China, e.g., Wang et al., AE 2015, doi: 10.1016/j.atmosenv.2015.04.063; Guo et al., GRL 2017, doi: 10.1002/2017GL073533; Liu et al., Sci. Rep. 2019, doi:10.1038/s41598-019-44284-2.

≫Lines 64-65: However, if aerosol concentration exceeds a certain threshold, this promoting effect disappears (Quan et al., 2011) or even turns into a suppressing effect due to the strong vapour competition (Koren et al., 2008; Rangognio, 2009).

Thank you for this valuable suggestion. We have added these references to the end of this sentence. Line 69: ......or even turns into a suppressing effect due to the strong vapour competition (Guo et al., 2017; Koren et al., 2008; Liu et al., 2019; Rangognio, 2009; Wang et al., 2015).

5. Lines 69-71: I notice that the work by Yan et al. JGR (2019) mentioned here is also from the same research group. Also, it occurs to me that the motivation seems a little confused: Since previous work has "quantitatively" proved…, why the authors attempt again to "quantitatively" confirm by model simulation of a fog event. Two "quantitatively " is redundant. Therefore, this sentence is suggested to be rephrased as follows: e.g. Our recent observational work (Yan et al., 2019) indicated a decreasing trend in fog days, and …"

≫Lines 69-71: Yan et al. (2019) analysed decadal trends of fog days and quantitatively proved that the inhibiting effects of urbanization outweigh the promoting effects of aerosols on fog during the mature urbanization stage. Their study inspires us to quantitatively confirm the roles of urbanization and aerosols......

Thank you for this valuable suggestion. The redundant "quantitatively" is deleted. We have changed this sentence to "Our recent observational work (Yan et al., 2019) indicated a decreasing trend in fog days, and the inhibiting effects of urbanization outweigh the promoting effects of aerosols on fog during the mature urbanization stage. This study aims to quantitatively confirm the roles of urbanization and aerosols......"(Line 73-76).

6. Line 75: "facilitates"-> "is expected to facilitate"

≫Line 75: This work facilitates the understanding of......

Thank you for this valuable suggestion. We have changed the following sentences: 1) Line 80: "This work is expected to facilitate the understanding of..."; 2) Line 306: "This study is expected to facilitate a better understanding of...".

7. Line 85: Something is suggested to be mentioned concerning Section 4 immediately after "Section 3.6 discusses the rationality and reliability of the results."

≫Line 85: Section 3.6 discusses the rationality and reliability of the results.

Thank you for this valuable suggestion. We have added "Section 4 concludes the findings of this study" to the end of this sentence (Line 90-91).

8. Line 90: it is suggested to clarify which city you are referring to? Since the reader cannot easily get any info from either text or Figure 1b.

≫Line 90: SX is ... approximately 30 km away from the nearest large city (Fig. 1b).

Thank you for this valuable suggestion. We have changed this sentence to "SX is ... approximately 30 km away from the nearest large city, Huainan (Fig. 1b)" (Line 96). The city of Huainan is also marked in Figure 1b.

9. Line 97: "replace"-> "used to replace"

≫Line 97: The data are resampled from 500 m to 30 arc-seconds (approximately 1 km) and replace the geological data of the WRF model.

Thank you for this valuable suggestion. We change this sentence to "The data are resampled from 500 m to 30 arc-seconds (approximately 1 km) and used to replace the geological data of the WRF model" (Line 103).

10. Lines 160-165: The logic seems a little problematic: since the fog holes are mainly caused by urbanization, as demonstrated in the references in this paragraph (aerosol effect is not mentioned and is supposed to not be the focus here), why you mentioned the effect of aerosol pollution. It is generally thought that urbanization effect tends to reduce LWP whereas aerosol tends to accumulate the formation of fog. The combined effect is highly dependent on the competing effect of the two factors. Here it is not accurate to argue that both of them "reducing the LWP or advancing the dissipation of fog".

≫Line 160-165: …… We assume that urbanization and air pollution could have profound effects on fog by reducing the LWP or advancing the dissipation of fog.

Thank you for this valuable suggestion. We agree that fog holes are mainly caused by urbanization, not by aerosols. We aimed to express that "the combined effects of urbanization and aerosols lead to fog holes", not "both of them lead to fog holes". To avoid the problem you mentioned, we change the last sentence to "We assume that urbanization could have profound effects on fog by reducing the LWP or advancing the dissipation of fog, and the role of aerosols on fog is weaker than that of urbanization" (Line 170-171).

11. Section 3.2: What are the criteria for you to determine a fog event from model-simuated LWP, which is required to be clarified here.

Thank you for this valuable suggestion. The criteria for fog is LWP>2 $g/m^2$ (Jia et al., 2018). We clarify it in Section 3.2: "Satellite cloud image and modelled LWP ($>2$ $g/m^2$) can represent the observed and simulated fog zone" (Line 173).

References:

Jia, X., Quan, J., Zheng, Z., Liu, X., Liu, Q., He, H., and Liu, Y.: Impacts of anthropogenic aerosols on fog in North China Plain, J. Geophys. Res.-Atmos., 124, 252–265, https://doi.org/10.1029/2018jd029437, 2018.

12. Section 3.3: The authors attempted to discuss the complicated non-monotonic effect of aerosol on fog formation by differing the emission rate, which is not very common. Why not used the aerosol concentration or CCN or Na that can well represent the real atmospheric pollution level for the time period investigated here? I am curious of the actual CCN or Na concentration for the experiment of u0e3?

Thank you for this valuable suggestion. We agree that CCN can better represent the air pollution level. The $CCN_{0.1}$ concentration of each experiment is marked in the new Figure 8 (Line 510-514), because the supersaturation in fog is commonly less than 0.1% (Mazoyer et al., 2016). The $CCN_{0.1}$ of current pollution level (u0e3) is 570 $cm^{-3}$.

[Figure]

Figure 8. Relationships of the microphysical parameters (LWC, $N_d$, $R_e$ and LWP) with emission level and $CCN_{0.1}$ concentrations. These parameters are the time-height averages (time average for the LWP) in fog.

References:

Mazoyer, M. , Burnet, F. , Roberts, G. C. , Haeffelin, M. , & Elias, T. (2016). Experimental study of the aerosol impact on fog microphysics. Atmospheric Chemistry and Physics, 1-35.

13. Lines 215-216: It will be misleading for the statement "the current pollution level in China is still located in the promoting regime rather than the suppressing regime of fog occurrence". Both ideal simulation (e.g., Rosenfeld et al. Science, 2008) or observational studies (Wang et al., AE 2015; Guo et al., GRL 2017) indicated that the tipping point tends to occur at AOD of 0.3-0.4 or CCN concentration of 1200/cm3. Recent observational work by Ilan Koren et al. (Science, 2014) suggested the cloud and precipitation is most sensitive to aerosol over the South Ocean. By comparison, the average AOD from MODIS in East China is on average much larger than 0.6, irrespective of the meteorological conditions.

≫Lines 215-216: The aerosol concentration of the transition point (experiment M2.5) is higher than that of u0e3 (Fig. 8), revealing that the current pollution level in China is still located in the promoting regime rather than the suppressing regime of fog occurrence.

Thank you for this valuable suggestion. The $CCN_{0.4}$ of u0e3 is 6023 $cm^{-3}$, higher than $CCN_{0.4}=1200$ $cm^{-3}$ that revealed by Rosenfeld et al. (Science, 2008). We agree that the AOD value of East China is larger than 0.6. It seems that the current pollution level could suppress fog rather than promotes fog.

The studies you listed mostly aim at convective clouds. Aerosols affect convective clouds through two competing mechanisms: 1) invigorating convection by promoting vapor condensation. 2) suppressing convection by blocking solar radiation and reducing surface heat flux. Under polluted conditions

(AOD>0.3 or $CCN_{0.4}$>1200 $cm^{-3}$), the suppressing effect outweighs the invigoration effect, so the turning point occurs (Koren et al. Science, 2008; Rosenfeld et al., Science, 2008). This suppressing effect does not exist in fog because fog commonly formed at night. Therefore, the turning point in fog might occur later than that in convective clouds. In North China Plain where air pollution is thought to be more serious, a case study by WRF-Chem also indicates that fog properties (e.g., LWC, $N_d$ and LWP) increase monotonically when emission intensity varies from 0.05-fold to 1-fold. It is consistent with our statement "the current pollution level in China is still located in the promoting regime rather than the suppressing regime of fog occurrence".

The above discussions have been included at the end of Section 3.4 (Line 224-232). Additionally, the statements are given by a more cautious manner: (Line 221-223): …possibly indicating that the current pollution level …; (Line 26, Abstract): the current pollution level in China could be still below the critical aerosol concentration that suppresses fog.

14. Lines 205-206: times is redundant and should be removed.

≫Lines 205-206: For example, the name M2.5 means multiplying by 2.5 times; the name D10 means dividing by 10 times.

Thank you for this valuable suggestion. We change this sentence to "For example, the name M2.5 means multiplying by 2.5; the name D10 means dividing by 10" (Line 211).

15. Line 448: it is better to indicate the year of 2017 following 03 January in the figure caption.

≫Line 448: Figure 2. The performance of the simulated fog zone at 08:00 03 January.

Thank you for this valuable suggestion. We change the caption to "The performance of the simulated fog zone at 08:00 03 January 2017" (Line 476).

16. Figure 3: it is suggested to show the major cities in the regions of interest shown in panel a and b, given the convenience to better understand the fog hole induced by urban. Besides, two subregions in Figure 3 is better to be marked in Figure 1 or 2.

Thank you for this valuable suggestion. The subregions of interest are marked in the new Figure 2 (Line 475-480). The cities with fog holes are marked in the new Figure 3 (Line 482-487).

[Figure]

Figure 2. The performance of the simulated fog zone at 08:00 03 January 2017. (a) Himawari 8 RGB composite cloud image overlaid with the MICAPS observation sites (light red dots) at which fog was observed (relative humidity > 90 % and VIS < 1 km). (b) Simulated LWP distribution. Only LWC below 1500 m are integrated. The blue dots are the SX site. The two dashed rectangles in (a) are the subregions of interest in Fig. 3.

[Figure]

Figure 3. Two sub-regions (a and b) with obvious fog holes on the Himawari 8 image at 11:00 03 January. The fog zone, which is represented by albedo > 0.45 (at 0.64 μm) and brightness temperature > 266 K (at 12.4 μm) (Di Vittorio et al., 2002), is marked with cold colours (blue or cyan). The urban areas are marked with blue or red. The red and white pixels surrounded or semi-surrounded by cold colours are fog holes, and among these pixels, the red pixels indicate the fog holes over urban areas. Some of the urbans with fog holes are marked by rectangles.

---

## Author Response (AR1)

Dear Referees,

Thanks for giving us an opportunity to revise our manuscript (acp-2019-1045). We appreciate your positive and constructive comments. We have studied these comments carefully and make revisions on the manuscript. These comments and the corresponding replies are listed below.

The referee's comments are highlighted by gray. The symbol "≫" quotes the original texts in the manuscript. Followed by the comments are our responses (normal texts) and current texts in the manuscript (leaded by **line number**). Some important revisions are colored by red. The revised manuscript with track changes are attached at the end of this file.

With regards,

Shuqi Yan, Bin Zhu*, and all co-authors.

**Replies to Referee#1**

**1.** Line 37-38: As indicated in IPCC AR5, "aerosol-cloud-radiation interactions" is suggested to be rephrased as "aerosol-radiation interaction" and "aerosol-cloud interaction" separately.

≫Line 37-38: ...which are called as aerosol-cloud-radiation interactions...

Thank you for this valuable suggestion. We have corrected it.

**Line 38-39 (Introduction)**

 ...which are called as aerosol-radiation and aerosol-cloud interactions...

**Line 458 (Table 1)**

 Aerosol-cloud-radiation interactions --> Aerosol-cloud and aerosol-radiation interactions

**2.** Line 41: "Many"->"Previous"

≫Line 41: Many studies have analysed...

Thank you for this valuable suggestion. We have changed all the "many studies" to "previous studies" (Line 35, 42, 219).

**3.** Line 45: "lower supersaturation" ??

≫Line 45: As a result, urban areas commonly experience higher temperatures and lower vapour contents. These conditions induce a lower supersaturation that is unfavourable for fog formation.

Thank you for this valuable suggestion. We have corrected this sentence.

**Line 46 (Introduction)**

…These conditions induce a lower relative humidity supersaturation that is unfavourable for fog formation.

**4.** Lines 64-65: Some important references are missing regarding the observational evidences of aerosol boomerang effect in China, e.g., Wang et al., AE 2015, doi: 10.1016/j.atmosenv.2015.04.063; Guo et al., GRL 2017, doi: 10.1002/2017GL073533; Liu et al., Sci. Rep. 2019, doi:10.1038/s41598-019-44284-2.

≫Lines 64-65: However, if aerosol concentration exceeds a certain threshold, this promoting effect disappears (Quan et al., 2011) or even turns into a suppressing effect due to the strong vapour competition (Koren et al., 2008; Rangognio, 2009).

Thank you for this valuable suggestion. We have added these references to the end of this sentence.

**Line 68 (Introduction)**

......or even turns into a suppressing effect due to the strong vapour competition (Guo et al., 2017; Koren et al., 2008; Liu et al., 2019; Rangognio, 2009; Wang et al., 2015).

**5.** Lines 69-71: I notice that the work by Yan et al. JGR (2019) mentioned here is also from the same research group. Also, it occurs to me that the motivation seems a little confused: Since previous work has "quantitatively" proved…, why the authors attempt again to "quantitatively" confirm by model simulation of a fog event. Two "quantitatively " is redundant. Therefore, this sentence is suggested to be rephrased as follows: e.g. Our recent observational work (Yan et al., 2019) indicated a decreasing trend in fog days, and …"

≫Lines 69-71: Yan et al. (2019) analysed decadal trends of fog days and quantitatively proved that the inhibiting effects of urbanization outweigh the promoting effects of aerosols on fog during the mature urbanization stage. Their study inspires us to quantitatively confirm the roles of urbanization and aerosols......

Thank you for this valuable suggestion. The redundant "quantitatively" is deleted. We have corrected this sentence.

**Line 72-76 (Introduction)**

Our recent observational work (Yan et al., 2019) indicated a decreasing trend in fog days, and the inhibiting effects of urbanization outweigh the promoting effects of aerosols on fog during the mature urbanization stage. This study aims to quantitatively confirm the roles of urbanization and aerosols......

**6.** Line 75: "facilitates"-> "is expected to facilitate"

≫Line 75: This work facilitates the understanding of......

Thank you for this valuable suggestion. We have corrected these sentences.

**Line 79 (Introduction)**

This work is expected to facilitate the understanding of...

**Line 305 (Conclusions)**

This study is expected to facilitate a better understanding of...

**7.** Line 85: Something is suggested to be mentioned concerning Section 4 immediately after "Section 3.6 discusses the rationality and reliability of the results."

≫Line 85: Section 3.6 discusses the rationality and reliability of the results.

Thank you for this valuable suggestion. We have added something after it.

**Line 89-90 (Introduction)**

Section 3.6 discusses the rationality and reliability of the results. Section 4 concludes the findings of this study.

**8.** Line 90: it is suggested to clarify which city you are referring to? Since the reader cannot easily get any info from
either text or Figure 1b.
≫Line 90: SX is ... approximately 30 km away from the nearest large city (Fig. 1b).

Thank you for this valuable suggestion. We have clarified this city, Huainan. It has been marked in Figure 1b.

**Line 95 (Section 2.1)**

SX is ... approximately 30 km away from the nearest large city, Huainan (Fig. 1b).

**9.** Line 97: "replace"-> "used to replace"
≫Line 97: The data are resampled from 500 m to 30 arc-seconds (approximately 1 km) and replace the geological data
of the WRF model.

Thank you for this valuable suggestion. We have corrected this sentence.

**Line 102 (Section 2.1)**

The data are resampled from 500 m to 30 arc-seconds (approximately 1 km) and used to replace the geological
data of the WRF model.

**10.** Lines 160-165: The logic seems a little problematic: since the fog holes are mainly caused by urbanization, as
demonstrated in the references in this paragraph (aerosol effect is not mentioned and is supposed to not be the focus
here), why you mentioned the effect of aerosol pollution. It is generally thought that urbanization effect tends to reduce

LWP whereas aerosol tends to accumulate the formation of fog. The combined effect is highly dependent on the competing effect of the two factors. Here it is not accurate to argue that both of them "reducing the LWP or advancing the dissipation of fog".

≫Line 160-165: …… We assume that urbanization and air pollution could have profound effects on fog by reducing the LWP or advancing the dissipation of fog.

Thank you for this valuable suggestion. We agree that fog holes are mainly caused by urbanization, not by aerosols. We aimed to express that "the combined effects of urbanization and aerosols lead to fog holes", not "both of them lead to fog holes". To avoid the problem you mentioned, we have corrected the last sentence.

**Line 169-170 (Section 3.1)**

We assume that urbanization could have profound effects on fog by reducing the LWP or advancing the dissipation of fog, and the role of aerosols on fog is weaker than that of urbanization.

**11.** Section 3.2: What are the criteria for you to determine a fog event from model-simuated LWP, which is required to be clarified here.

Thank you for this valuable suggestion. The criteria for fog is LWP$>2$ g/m$^2$ (Jia et al., 2018). We have clarified it in Section 3.2.

**Line 173 (Section 3.2)**

Satellite cloud image and modelled LWP ($>2$ g/m$^2$) can represent the observed and simulated fog zone (Jia et al., 2018).

**References**

Jia, X., Quan, J., Zheng, Z., Liu, X., Liu, Q., He, H., and Liu, Y.: Impacts of anthropogenic aerosols on fog in North China Plain, J. Geophys. Res.-Atmos., 124, 252–265, https://doi.org/10.1029/2018jd029437, 2018.

**12.** Section 3.3: The authors attempted to discuss the complicated non-monotonic effect of aerosol on fog formation by
differing the emission rate, which is not very common. Why not used the aerosol concentration or CCN or Na that can
well represent the real atmospheric pollution level for the time period investigated here? I am curious of the actual
CCN or Na concentration for the experiment of u0e3?

Thank you for this valuable suggestion. We agree that CCN can better represent the air pollution level. The $CCN_{0.1}$
concentration of each experiment is marked in the new Figure 8, because the supersaturation in fog is commonly less
than 0.1% (Mazoyer et al., 2016). The $CCN_{0.1}$ of current pollution level (u0e3) is 570 $cm^{-3}$.

**Line 510-514 (Figure 8)**

[Figure]

Figure 8. Relationships of the microphysical parameters (LWC, $N_d$, $R_e$ and LWP) with emission level and $CCN_{0.1}$ con-
centrations. These parameters are the time-height averages (time average for the LWP) in fog.

[Figure]

Figure 2. The performance of the simulated fog zone at 08:00 03 January 2017. (a) Himawari 8 RGB composite cloud
image overlaid with the MICAPS observation sites (light red dots) at which fog was observed (relative humidity > 90 %
and VIS < 1 km). (b) Simulated LWP distribution. Only LWC below 1500 m are integrated. The blue dots are the SX
site. The two dashed rectangles in (a) are the subregions of interest in Fig. 3.

**Line 480-486 (Figure 3)**

[Figure]

Figure 3. Two sub-regions (a and b) with obvious fog holes on the Himawari 8 image at 11:00 03 January 2017. The fog zone, which is represented by albedo > 0.45 (at 0.64 μm) and brightness temperature > 266 K (at 12.4 μm) (Di Vit- torio et al., 2002), is marked with cold colours (blue or cyan). The urban areas are marked with blue or red. The red and white pixels surrounded or semi-surrounded by cold colours are fog holes, and among these pixels, the red pixels indicate the fog holes over urban areas. Some of the cities with fog holes are marked by rectangles.

**Replies to Referee#2**

We appreciate your valuable suggestions of our manuscript.

**Replies to Referee#3**

**1.** The focus of this study is on the radiative fog. However, there are different fog formation in atmosphere. For example, the advection fog formation is often occurred in the coast of eastern China. The Authors should highlight that under different fog conditions (i.e., radiative fog or advection fog, etc.) what is the effects of the urbanization and aerosol particles on the fog formation.

Thank you for this valuable suggestion. We agree that radiation fog and advection fog are two major fog types in China. They can occur in both inland and coastal areas. Gu et al. (2019) revealed the occurring frequencies of different fog types in Shanghai, a coastal city, during the past three decades. The major fog type is radiation fog (38.3%), followed by advection fog (27.7%) and advection-radiation fog (23.4%). Therefore, we infer that the dominant fog type in inland areas and coastal areas is radiation fog, which should be attracted more attention.

Compared with radiation fog which usually occurs under stagnant weather conditions, advection fog is associated with synoptic forcing, i.e., advection of a moist air mass with contrasting temperature properties with respect to the underlying surface (Gultepe et al., 2007). The role of synoptic forcing should be considered when studying the effects of urbanization and aerosols on advection fog, which is more complex than radiation fog. Zhong et al. (2017) indicated that urbanization and aerosols have nonsignificant effects on convective precipitation when the synoptic forcing is strong. Therefore, this study focuses on radiation fog to study the effects of urbanization and aerosols.

[revised manuscript text omitted]

批注 [yansq23]: $CCN_{0.1}$ is marked under the corresponding experiments

批注 [yansq24]: Referee#1_Comment12 & 13

[Figure]

Figure 9. Similar to Fig. 5, but for the combined effect of urbanization and aerosols (u3e3 minus u0e0).

[Figure]

Figure 10. The combined effect of urbanization and aerosols (u3e3 minus u0e0) on various items of the LWC budget. The three rows are the hourly tendencies (g kg$^{-1}$) of the microphysical, boundary layer, and advection processes.

[Figure]

Figure 11. The combined effect of urbanization and aerosols (u3e3 minus u0e0) on various items of the microphysical tendency. The three rows are the  hourly tendencies (g kg$^{-1}$) of the microphysical, condensation/evaporation, and sedimentation processes.

---

## Author Response (AR2)

Dear Editor,

Thanks for giving us an opportunity to revise our manuscript (acp-2019-1045). We appreciate your positive and con- structive comments. We have studied these comments carefully and make revisions on the manuscript. These com- ments and the corresponding replies are listed below.

The comments are highlighted by gray. The symbol "≫" quotes the original texts in the manuscript. Followed by the comments are our responses and revisions in the manuscript. Some important revisions are colored by red. The tracked change is attached at the end of this file.

With regards,

Shuqi Yan, Bin Zhu*, and all co-authors.

**1.** You need to emphasize what is the urbanization effect. You mention land use change in your model sensitivity ex- periments. Does urbanization change the surface albedo, surface roughness, surface flux, ...etc? it need to be clearly stated.

≫Line 82: ... urbanization mainly refers to UHI and UDI associated with land use change and human activities…

Thank you for this valuable suggestion. The urbanization effect here refers to UHI and UDI induced by anthropogenic heating and the land use change. The land use change includes the changes in corresponding surface properties, e.g., surface albedo, surface roughness, surface flux.

**Revision in line 82 (Introduction)**

urbanization mainly refers to UHI and UDI  induced by an- thropogenic heating and land use change with the corresponding surface property change (e.g., surface albedo, surface roughness, surface flux).

**2.** Line 43, you say "The urban surface has a lower albedo". compared to what? to rural surface? why?.

Thank you for this valuable suggestion. We compare urban surface with rural surface. The albedo is 0.15 for urban surface and 0.20 for rural surface in the WRF model setting.

**Revision in line 44 (Introduction)**

The urban surface has a lower albedo than do rural surface.

**3.** Line 60-61. "The aerosol indirect effect on clouds..." but here in the context you are talking about aerosol effect on fog, not on clouds.

≫Line 60-61. The aerosol indirect effect on cloud is addressed as one of the most uncertain factors in the IPCC report. Aerosol concentration has a two-fold effect on fog, which is called as the boomerang pattern (Koren et al., 2008).

Thank you for this valuable suggestion. We have corrected this sentence.

**Revision in line 60-63 (Introduction)**

The aerosol indirect effect on cloud is addressed as one of the most uncertain factors in the IPCC report. This effect on fog is also complex and two-fold, which is determined by aerosol concentration.

**4.** WRF-Chem. Please give the full name when it is first time mentioned in the paper.

Thank you for this valuable suggestion. We have given the full name "Weather Research and Forecasting with Chemistry" in Abstract and Introduction.

**Revision in line 20 (Abstract)**

…a dense radiation fog event in East China in January 2017 was reproduced by the Weather Research and Forecasting with Chemistry (WRF-Chem) model.

**Revision in line 76 (Introduction)**

…by an online-coupled synoptic and air quality model, Weather Research and Forecasting with Chemistry
(WRF-Chem).

**5.** Line 151 "figure not show". I'd like you to show the figure and put it in supplement.

≫Lines 151: During fog period (Fig. 4 shaded zone), the three methods nearly yield the same results (figure not
shown).

Thank you for this valuable suggestion. We did not show the visibility calculated by the three parameterization meth-
ods (corresponding to Equations 1 to 3). We have added Fig. S1 in the supplement.

[Figure]

Figure S1. Comparisons of VIS calculated by Equations 1, 2 and 3. The fog period (observed VIS < 1 km and RH >
90 %) is shaded with light yellow. Note that Equations 1 and 2 only consider the extinction by fog water, while Equa-
tion 3 considers the extinction by fog water and aerosols.

**6.** Line 168. "We assume that urbanization could have..." I'd change the sentence to something like "We hypothesize that ..."

Thank you for this valuable suggestion. We have corrected this sentence.

**Revision in line 170 (Section 3.1.2)**

We hypothesize that urbanization could have….

**7.** Line 207. The first time "boomerang pattern" occurs please explain it. I know that you explain it later.

Thank you for this valuable suggestion. We think that the term "boomerang pattern" is not very common. In case the readers might get confused, we avoid this term and change the corresponding expressions.

**Revision in line 61-63 (Introduction)**

The aerosol indirect effect on cloud is addressed as one of the most uncertain factors in the IPCC report. This effect on fog is also complex and two-fold, which is determined by aerosol concentration…….

**Revision in line 209 (Section 3.4)**

It is probable that the current pollution level of China always promotes fog occurrence. To testify whether the u0e3 is below the transition point  that suppresses fog...

**Revision in line 214 (Section 3.4)**

 The variation shape of the four parameters  demonstrates that the model is able to simulate the dual effects of aerosols.

**Revision in line 293 (Conclusions)**

Further sensitivity experiments show that the current pollution level in China could be still below the  critical aerosol concentration that suppresses fog.

**8.** Line 244. "boundary layer and advection tendencies is equal to the LWC distribution". The sentence is not correct. how can tendencies equal to state (LWC). tendencies are changes of state (e.g., LWC).

≫Line 244: The sum of microphysical (condensation/evaporation and sedimentation), boundary layer and advection tendencies is equal to the LWC distribution.

Thank you for this valuable suggestion. The tendencies are changing rates ($\frac{\Delta \text{LWC}}{\Delta t}$) in unit of g/kg h$^{-1}$. We aim to express that summing the integral of these tendencies with respect to time ($\sum \frac{\Delta \text{LWC}}{\Delta t} \Delta t$) equals to LWC, where $\Delta t$ is the model output time interval (1 hour).

**Revision in line 245-247 (Section 3.5)**

Summing the integral of microphysical (condensation/evaporation and sedimentation), boundary layer and advection tendencies with respect to time equals to the LWC distribution.

**9.** Figures 10 and 11. here the unit of tendencies should be g/kg/time.

Thank you for this valuable suggestion. We have changed the unit to be g kg$^{-1}$ h$^{-1}$ in Fig. 10, Fig. 11, Fig. S3 and Fig. S4.

[revised manuscript text omitted]

---

## Author Response (AR3)

Dear Editor,

Thanks for giving us an opportunity to revise our manuscript (acp-2019-1045). We appreciate your positive and con-
structive comments. We have checked the technical errors carefully and make revisions on the manuscript. These
comments and the corresponding replies are listed below.

The comments are highlighted by gray. The symbol "≫" quotes the original texts in the manuscript. Followed by the
comments are our responses and revisions in the manuscript. Some important revisions are colored by red. The tracked
change is attached at the end of this file.

[revised manuscript text omitted]

**WPS domain**

(a)

36° N

33° N

30° N

27° N

113° E    116° E    119° E    122° E

**Height (m)**

1000    2000    3000    4000    5000    6000

**d02 land use**

33.5° N (b)

33° N

32.5° N

32° N

31.5° N

116° E  116.5° E  117° E  117.5° E  118° E

**Land use type**

2  3  4  5  6  7  8  9 10 11 12 13 14 15 16 17 18 19 20

| 1 evergreen needleleaf | 2 evergreen broadleaf | 3 deciduous needleleaf | 4 deciduous broadleaf | 5 mixed forests |
|---|---|---|---|---|
| 6 closed shrubland | 7 open shrublands | 8 woody savannas | 9 savannas | 10 grasslands |
| 11 permanent wetlands | 12 croplands | 13 urban and built-up | 14 cropland vegetation mosaic | 15 snow and ice |
| 16 barren or sparse | 17 water | 18 wooded tundra | 19 mixed tundra | 20 barren tundra |

Figure 1. (a) The WRF domain overlaid with terrain height. (b) The land use distribution of domain d02. The green dot
is Hefei, the capital of Anhui Province. The white dot is Huainan. The two red dots are the SX site. The land use and
emissions of the 22 km × 26 km black box in the center of (b) will be altered in the sensitivity experiments.

[Figure]

Figure 2. The performance of the simulated fog zone at 08:00 03 January 2017. (a) Himawari 8 RGB composite cloud
image overlaid with the MICAPS observation sites (light red dots) at which fog was observed (relative humidity > 90 %
and VIS < 1 km). (b) Simulated LWP distribution. Only LWC below 1500 m are integrated. The blue dots are the SX
site. The two dashed rectangles in (a) are the subregions of interest in Fig. 3.

[Figure]

Figure 3. Two sub-regions (a and b) with obvious fog holes on the Himawari 8 image at 11:00 03 January 2017. The fog zone, which is represented by albedo > 0.45 (at 0.64 μm) and brightness temperature > 266 K (at 12.4 μm) (Di Vittorio et al., 2002), is marked with cold colours (blue or cyan). The urban areas are marked with blue or red. The red and white pixels surrounded or semi-surrounded by cold colours are fog holes, and among these pixels, the red pixels indicate the fog holes over urban areas. Some of the cities with fog holes are marked by rectangles.

[Figure]

Figure 4. The performance of the simulated meteorological parameters at the SX site. (a) VIS. (b) air temperature. (c) 10-minute average wind speed. (d) Relative humidity (RH). The red dotted lines represent the model results, and the black lines are the observations. The fog period (VIS < 1 km and RH > 90 %) is shaded with light yellow.

[Figure]

Figure 5. Time-height distribution of the LWC (g kg$^{-1}$) in (a) u0e0 and (b) u3e0, and (c) is the urbanization effect (u3e0 minus u0e0) on LWC. The two white curves in (c) are the LWP. The black contour lines in (c) are the difference of vertical velocity (cm s$^{-1}$) (u3e0 minus u0e0). Only the lines after 00:00 are shown for clarity.

[Figure]

Figure 6. Profiles of the LWC (first row), temperature (Tem) (f, g, j) and vertical vapour flux divergence (VFD) (h, i)

(g h$^{-1}$ m$^{-2}$·hpa$^{-1}$) in u0e0 and u3e0 at different times.

[Figure]

Figure 7. Similar to Fig. 5, but for the aerosol effect (u0e3 minus u0e0).

[Figure]

Figure 8. Relationships of the microphysical parameters (LWC, $N_d$, $R_e$ and LWP) with emission level and $CCN_{0.1}$ concentrations. These parameters are the time-height averages (time average for the LWP) in fog.

[Figure]

Figure 9. Similar to Fig. 5, but for the combined effect of urbanization and aerosols (u3e3 minus u0e0).

[Figure]

Figure 10. The combined effect of urbanization and aerosols (u3e3 minus u0e0) on various items of the LWC budget.

The three rows are the tendencies (g kg$^{-1}$ h$^{-1}$) of the microphysical, boundary layer, and advection processes.

[Figure]

Figure 11. The combined effect of urbanization and aerosols (u3e3 minus u0e0) on various items of the microphysical tendency. The three rows are the tendencies (g kg⁻¹ h⁻¹) of the microphysical, condensation/evaporation, and sedimentation processes.